# SAMPLE-SPECIFIC NOISE INJECTION FOR DIFFUSION-BASED ADVERSARIAL PURIFICATION

## ABSTRACT

*Diffusion-based purification* (DBP) methods aim to remove adversarial noise from the input sample by first injecting Gaussian noise through a forward diffusion process, and then recovering the clean example through a reverse generative process. In the above process, how much Gaussian noise is injected to the input sample is key to the success of DBP methods, which is controlled by a constant noise level $t^*$ for all samples in existing methods. In this paper, we discover that an optimal $t^*$ for each sample indeed could be different. Intuitively, the cleaner a sample is, the less the noise it should be injected, and vice versa. Motivated by this finding, we propose a new framework, called *Sample-specific Score-aware Noise Injection* (SSNI). Specifically, SSNI uses a pre-trained score network to estimate how much a data point deviates from the clean data distribution (i.e., score norms). Then, based on the magnitude of score norms, SSNI applies a reweighting function to adaptively adjust $t^*$ for each sample, achieving sample-specific noise injections. Empirically, incorporating our framework with existing DBP methods results in a notable improvement in both accuracy and robustness on CIFAR-10 and ImageNet-1K, highlighting the necessity to allocate *distinct noise levels to different samples* in DBP methods. Our code is available at: https://anonymous.4open.science/r/SSNI-F746.

## 1 INTRODUCTION

*Deep neural networks* (DNNs) are vulnerable to adversarial examples, which is a longstanding problem in deep learning (Szegedy et al., 2014; Goodfellow et al., 2015). Adversarial examples aim to mislead DNNs into making erroneous predictions by adding imperceptible adversarial noise to clean examples, which pose a significant security threat in critical applications (Dong et al., 2019; Finlayson et al., 2019; Cao et al., 2021; Jing et al., 2021). To defend against adversarial examples, *adversarial purification* (AP) stands out as a representative defensive mechanism, by leveraging pre-trained generative models to purify adversarial examples back towards their natural counterparts before feeding into a pre-trained classifier (Yoon et al., 2021; Nie et al., 2022). Notably, AP methods benefit from their modularity, as the purifier operates independently of the downstream classifier, which facilitates seamless integration into existing systems and positions AP as a practical approach to improve the adversarial robustness of DNN-based classifiers.

Recently, *diffusion-based purification* (DBP) methods have gained much attention as a promising framework in AP, which leverage the denoising nature of diffusion models to mitigate adversarial noise (Nie et al., 2022; Xiao et al., 2023; Lee & Kim, 2023). Generally, diffusion models train a forward process that maps from data distributions to simple distributions, e.g., Gaussian, and reverse this mapping via a reverse generative process (Ho et al., 2020; Song et al., 2021b). When applied in DBP methods, the forward process gradually injects Gaussian noise into the input sample, while the reverse process gradually purify noisy sample to recover the clean sample. The quality of the purified sample heavily depends on the amount of Gaussian noise added to the input during the forward process, which can be controlled by a noise level parameter $t^*$. Existing DBP methods (Nie et al., 2022; Xiao et al., 2023; Lee & Kim, 2023) manually select a constant $t^*$ for all samples.

However, we find that using a sample-shared $t^*$ may *overlook* the fact that an optimal $t^*$ indeed could be different at sample-level, as demonstrated in Figure 1. For example, in Figure 1a, $t^* = 100$ is too small, resulting in the adversarial noise not being sufficiently removed by the diffusion models.

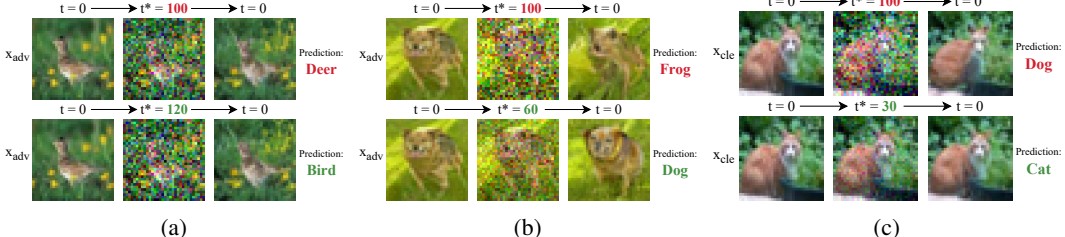

(a)                              (b)                              (c)

Figure 1: For each sub-figure: the 1st column contains the input (i.e., could either be AEs or CEs), the 2nd column contains noise-injected examples with different $t^*$s, and the 3rd column contains purified examples. We use *DiffPure* (Nie et al., 2022) with a sample-shared $t^* = 100$ selected by Nie et al. (2022) to conduct this experiment on CIFAR-10 (Krizhevsky et al., 2009). The globally shared $t^* = 100$ offers a baseline, but results in suboptimal prediction performance compared to what could be achieved by tuning the noise level for individual samples. Notably, while the recovered images obtained by different noise levels may be visually indistinguishable, they carry different semantics. For instance, the image is classified as "frog" (incorrect) with $t^* = 100$ but as "dog" (correct) with $t^* = 60$ (Figure 1b). These *highlight the need for a sample-wise noise level adjustment*.

This is because diffusion models are good at denoising samples that have been sufficiently corrupted by Gaussian noise through the forward process (Ho et al., 2020; Song et al., 2021b). With a small $t^*$, the sample remains insufficiently corrupted, which limits the denoising capability in the reverse process and thereby compromising the robustness against adversarial examples. On the other hand, in Figure 1b and 1c, $t^* = 100$ is too large, resulting in excessive disruption of the sample's semantic information during the forward process, which makes it difficult to recover the original semantics in the reverse process. In this case, both robustness and clean accuracy are compromised, as the purified samples struggle to preserve the semantic consistency of clean samples. These observations motivate us to make the *first* attempt to adjust the noise level on a sample-specific basis.

In this paper, we propose *Sample-specific Score-aware Noise Injection* (SSNI), a new framework that leverages the distance of a sample from the clean data distribution to adaptively adjust $t^*$ on a sample-specific basis. SSNI aims to inject less noise to cleaner samples, and vice versa.

To implement SSNI, inspired by the fact that *scores* (i.e., $\nabla_{\mathbf{x}} \log p_t(\mathbf{x})$) reflect the directional momentum of samples toward the high-density areas of clean data distribution (Song & Ermon, 2019), we use *score norms* (i.e., $\|\nabla_{\mathbf{x}} \log p_t(\mathbf{x})\|$) as a natural metric to measure the deviation of a data point from the clean data distribution. In Section 3, we establish the relationship between the score norm and the noise level required for different samples. Specifically, samples with different score norms tend to have accumulated different noise levels. Furthermore, we empirically show that the cleaner samples – those closer to the clean data distribution – exhibit lower score norm, justifying the rationale of using score norms for reweighting $t^*$. Concretely, we use a pre-trained score network to estimate the score norm for each sample. Based on this, we propose two reweighting functions that adaptively adjust $t^*$ according to its score norm, achieving sample-specific noise injections (see Section 4.3). Notably, this reweighting process is *lightweight*, ensuring that SSNI is computationally feasible and can be applied in practice with minimal overhead (see Section 5.5).

Through extensive evaluations on benchmark image datasets such as CIFAR-10 (Krizhevsky et al., 2009) and ImageNet-1K (Deng et al., 2009), we demonstrate the effectiveness of SSNI in Section 5. Specifically, combined with different DBP methods (Nie et al., 2022; Xiao et al., 2023; Lee & Kim, 2023), SSNI can boost clean accuracy and robust accuracy *simultaneously* by a notable margin against the well-designed adaptive white-box attack (see Section 5.2 and Algorithm 2).

The success of SSNI takes root in the following aspects: (1) an optimal noise level $t^*$ for each sample indeed could be different, making SSNI a more effective approach to unleash the intrinsic strength of DBP methods; (2) existing DBP methods often inject excessive noise into clean samples, resulting in a degradation in clean accuracy. By contrast, SSNI injects less noise to clean samples, and thereby notably improving the clean accuracy. Meanwhile, SSNI can effectively handle adversarial samples by injecting sufficient noise on each sample; (3) SSNI is designed as general framework instead of a specific method, allowing it to be seamlessly integrated with a variety of existing DBP methods.

## 2 PRELIMINARY AND RELATED WORK

In this section, we first review the concepts of diffusion models and scores in detail. Then, we review the related work of DBP methods.

**Diffusion models** are generative models designed to approximate the underlying clean data distribution $p(\mathbf{x}_0)$, by learning a parametric distribution $p_\theta(\mathbf{x}_0)$ with a *forward* and a *reverse* process. In *Denoising Diffusion Probabilistic Models* (DDPM) (Ho et al., 2020), the *forward* process of a diffusion model defined on $\mathcal{X} \subseteq \mathbb{R}^d$ can be expressed by:

$$q(\mathbf{x}_t|\mathbf{x}_0) = \mathcal{N}\left(\mathbf{x}_t; \sqrt{\bar{\alpha}}\mathbf{x}_0, (1 - \bar{\alpha}_t)\mathbf{I}\right), \tag{1}$$

where $\bar{\alpha}_t = \prod_{i=1}^t (1 - \beta_i)$ and $\{\beta_t\}_{t \in [0,T]}$ are predefined noise scales with $\beta_t \in (0, 1)$ for all $t$. As $t$ increases, $\mathbf{x}_t$ converges toward isotropic Gaussian noise. In the *reverse* process, DDPM seeks to recover the clean data from noise by simulating a Markov chain in the reverse direction over $T$ steps. The reverse transition at each intermediate step is modeled by

$$p_\theta(\mathbf{x}_{t-1}|\mathbf{x}_t) = \mathcal{N}(\mathbf{x}_{t-1}; \mu_\theta(\mathbf{x}_t, t), \sigma_t^2\mathbf{I}), \tag{2}$$

where $\mu_\theta(\mathbf{x}_t, t) = \frac{1}{\sqrt{1-\beta_t}}\left(\mathbf{x}_t - \frac{\beta_t}{\sqrt{1-\bar{\alpha}_t}}\boldsymbol{\epsilon}_\theta(\mathbf{x}_t, t)\right)$ is the predicted mean at $t$, and $\sigma_t$ is a fixed variance (Ho et al., 2020). More specifically, the model predicts the noise $\boldsymbol{\epsilon}_{\theta^*}(\mathbf{x}_t, t)$ using a U-net architecture, allowing the estimation of $\mu_\theta(\mathbf{x}_t, t)$ at each noise level. The training objective minimizes the distance between the true noise and the predicted noise:

$$\boldsymbol{\theta}^* = \arg\min_{\boldsymbol{\theta}} \mathbb{E}_{\mathbf{x}_0, t, \boldsymbol{\epsilon}}\left[\left\|\boldsymbol{\epsilon} - \boldsymbol{\epsilon}_{\boldsymbol{\theta}}\left(\sqrt{\bar{\alpha}_t}\mathbf{x}_0 + \sqrt{1 - \bar{\alpha}_t}\boldsymbol{\epsilon}, t\right)\right\|_2^2\right].$$

During inference, starting from $\mathbf{x}_T \sim \mathcal{N}(\mathbf{0}, \mathbf{I})$, the model generates a sample by iteratively sampling $\mathbf{x}_{t-1}$ from $\mathbf{x}_t$ using the learned reverse process over all $1 \leq t \leq T$:

$$\hat{\mathbf{x}}_{t-1} = \frac{1}{\sqrt{1-\beta_t}}\left(\hat{\mathbf{x}}_t - \frac{\beta_t}{\sqrt{1-\bar{\alpha}_t}}\boldsymbol{\epsilon}_{\theta^*}(\hat{\mathbf{x}}_t, t)\right) + \sqrt{\beta_t}\boldsymbol{\epsilon}, \quad \text{with } \boldsymbol{\epsilon} \sim \mathcal{N}(\mathbf{0}, \mathbf{I}). \tag{3}$$

**Score and score norm.** In this paper, a *score* refers to the gradient of the log-probability density, i.e., $\nabla_{\mathbf{x}}\log p_t(\mathbf{x})$, which represents the direction of maximum increase in the log-density across a vector field (Song & Ermon, 2019). The *score norm* is the magnitude (or length) of this gradient, denoted as $\|\nabla_{\mathbf{x}}\log p(\mathbf{x})\|$. It reflects how much a data point deviates from the clean data distribution. A larger score norm suggests the data point is situated in a low-probability region, while a smaller score norm indicates it is closer to regions where clean data occurs. Yoon et al. (2021) discover that score norms can effectively differentiate adversarial examples from clean examples. Building on this, Zhang et al. (2023) propose a more robust score called the *expected perturbation score* (EPS), which computes the expected scores of perturbed samples over a range of noise levels. EPS improves robustness against noise variations, providing a more reliable metric for detecting adversarial samples.

**Diffusion-based purification.** Adversarial purification (AP) leverages generative models as an add-on module to purify adversarial examples before classification. With in this context, *diffusion-based purification* (DBP) methods have emerged as a promising framework, exploiting the inherent denoising nature of diffusion models to filter out adversarial noise (Nie et al., 2022; Wang et al., 2022; Xiao et al., 2023; Lee & Kim, 2023). Specifically, Nie et al. (2022) and Xiao et al. (2023) integrate diffusion models to purify adversarial inputs. Wang et al. (2022) introduce input guidance during the reverse diffusion process to ensure the purified outputs stay close to the inputs. Lee & Kim (2023) further establish a reliable evaluation framework for DBP methods and propose a fine-tuned gradual noise scheduling for multi-step purifications. More recently, Bai et al. (2024) propose to guide diffusion models with contrastive loss through the reverse process.

## 3 MOTIVATION

In this section, we elaborate on the motivation of our method by connecting the impact of different perturbation budgets $\epsilon$ to the required noise level $t^*$ of each sample through score norm.

**Sample-shared noise-level $t^*$ fails to address diverse perturbation budgets.** We empirically observe that an optimal noise level $t^*$ for each sample indeed could be different. Figure 1 illustrates

that, while a constant noise level $t^* = 100$, as suggested by Nie et al. (2022), yields strong performances for some samples, it leads to *suboptimal* results for others. Since DBP relies on adequate $t^*$ for forward noise injection to remove adversarial perturbation, a shared $t^*$ cannot adapt to the distinct perturbation budgets of individual examples, leading to a suboptimal accuracy-robustness trade-off. Specifically, Figure 1a shows that $t^* = 100$ is insufficient remove the adversarial noise, leaving residual perturbations that compromise robustness. On the contrary, Figure 1c and 1b reveal that $t^* = 100$ overly suppresses the other sample's semantic information during the forward process, making it difficult to recover the original semantics via the reverse process. These findings *highlight the need for sample-specific noise injection levels tailored to individual perturbation budgets*.

**Score norms vary across perturbation budgets.** Adversarial and clean examples are from distinct distributions (Gao et al., 2021). Motivated by this, we further investigate how different perturbation budgets $\epsilon$ affect score norms under adversarial attacks (see Appendix B). Specifically, we compute the score norm of different samples undergoing PGD+EOT $\ell_\infty(\epsilon = 8/255)$ with perturbation budgets varying between 0 and $8/255$ on CIFAR-10. Our finding reveals a consistent pattern: The results consistently reveal that *samples subjected to stronger perturbations (higher $\epsilon$) exhibit larger score norms, whereas cleaner samples (lower $\epsilon$) show smaller score norms.* These results imply the utility of score norms can be extended from detecting adversarial samples from clean ones, as shown by (Yoon et al., 2021), to *differentiate adversarial examples based on their perturbation strength.*

**Different score norms imply different $t^*$.** Building on the observed relationship between perturbation budgets and score norms, we next explore how score norms of different samples correlate with the noise level $t^*$, hypothesizing that *score norms can inform noise strengths $t^*$ tailored to individual samples*. The proofs of the following Lemma 1 and Proposition 1 are in Appendix A.

We consider a diffusion model over a measurable space $\mathcal{X} \subseteq \mathbb{R}^d$, with forward process characterized by a Gaussian transition kernel $q(\mathbf{x}_t|\mathbf{x}_0)$ as defined in Eq. (1).

**Lemma 1.** *Suppose there exists a constant $K > 0$ such that for all $t \geq 0$ and all $\mathbf{x}_t \in \mathcal{X}$, the expected norm of $\mathbf{x}_0$ given $\mathbf{x}_t$ satisfies: $\mathbb{E}_{\mathbf{x}_0 \sim p(\mathbf{x}_0|\mathbf{x})}[\|\mathbf{x}_0\|] \leq K \|\mathbf{x}_t\|$. Then, there exist constants $0 < C < 1$ and $T_0 > 0$ such that for all $t \geq T_0$:*

$$\|\nabla_\mathbf{x} \log p_t(\mathbf{x})\| > C \|\mathbf{x}\|.$$

Lemma 1 establishes that we can find a time threshold $T_0$, after which the score norm maintains proportional to the input norm as $t$ increases (and higher noise levels $t^*$ as diffusion steps accumulate). Building on this, we now investigate how the score norm varies between different time steps $t_1$ and $t_2$, providing a lower bound on the difference in score norms over time.

**Proposition 1.** *Consider the diffusion model satisfying all conditions as specified in Lemma 1. Assume that there exist constants $K > 0$, such that $\beta_t \leq K$ for all $t \geq 0$. Additionally, suppose $\|\mathbf{x}\| \leq M$ for any $\mathbf{x} \in \mathcal{X}$, for some $M > 0$. Then, for any $\epsilon$, there exists a constant $\Delta = 2\epsilon/(CK)$ such that for $t_1, t_2 \geq 0$, we have:*

$$|\|\nabla_\mathbf{x} \log p_{t_1}(\mathbf{x})\| - \|\nabla_\mathbf{x} \log p_{t_2}(\mathbf{x})\|| > \epsilon, \ \ \text{with } |t_1 - t_2| \geq \Delta.$$

Proposition 1 shows that the score norm varies with $t$, and the variation exceeds a threshold $\epsilon > 0$ when the time difference $|t_2 - t_1|$ is sufficiently large. Consequently, for two noisy samples $\boldsymbol{x}_1$ and $\boldsymbol{x}_2$ with different score norms, the monotonic increase of $\{\beta_t\}_{t\in[0,T]}$ with $t$ implies that different score norms correspond to different noise levels, i.e., $t_1 \neq t_2$.

Recall that DBP involves Gaussian noise injection (i.e., forward process) followed by denoising in the reverse process to remove the adversarial perturbation. As samples $\boldsymbol{x}_1$ and $\boldsymbol{x}_2$ differ in score norms, they inherently present different denoising difficulties and require different optimal noise levels $t_1^* \neq t_2^*$ accordingly. This aligns with the established intuition that score norms reflect perturbation strengths and corresponding denoising requirements.

Motivated by the relationship between $\epsilon$, score norms, and $t^*$, we propose using *score norm as an indicator of denoising difficulty* to reweight the noise injection level, which transforms the globally-shared noise level into sample-specific configurations, tailoring the denoising strength to each sample's specific requirements. The success of this hypothesis is further empirically validated by the purification outcomes reported in Sec. 5.

*Remark* 1 (Limitation). We acknowledge that Proposition 1 does not explicitly focus on the properties of adversarial attacks, which is often infeasible to analyze rigorously. To simplify the setup, we

Figure 2: An overview of the proposed SSNI framework. Compared to existing DBP methods, SSNI use a pre-trained score network $s_\theta$ to estimate the score norm $\|s_\theta(\mathbf{x})\|$. Then, SSNI applies a reweighting function $f$ to adaptive adjust $t^*$ for each sample $\mathbf{x}_i$ based on its score norm $\|s_\theta(\mathbf{x}_i)\|$, achieving sample-specific noise injections. Notably, SSNI is designed as general framework instead of a specific method, allowing it to be seamlessly integrated with a variety of existing DBP methods.

disregard potential compounding interactions between adversarial perturbation and injected Gaussian noise in the forward pass of DBP. Instead, we leverage the properties of diffusion models and present the analysis as a *conceptual motivation to inspire the design of the proposed method*, rather than as a basis for a comprehensive theory.

## 4 SAMPLE-SPECIFIC SCORE-AWARE NOISE INJECTION

Motivated by Section 3, we propose *Sample-specific Score-aware Noise Injection* (SSNI), a flexible framework that adaptively adjusts the noise level for each sample based on how much it deviates from the clean data distribution, as measured by the score norm. We begin by introducing the SSNI framework, followed by a connection with related work and the empirical realization of SSNI.

### 4.1 FRAMEWORK OF SSNI

**Overview**. SSNI builds upon existing DBP methods by reweighting the optimal noise level $t^*$ from a global, sample-shared constant to a sample-specific quantity. The core idea behind SSNI is to leverage score norms to modulate the noise injected to each sample during the diffusion process, ensuring a more targeted denoising process tailored to each individual sample. For clarity, we provide a visual illustration of SSNI in Figure 2, and the procedure is described in Algorithm 1.

**DBP with sample-shared noise level** $t^*$. Existing DBP methods use an off-the-shelf diffusion model for data purification, and a classifier responsible for label prediction. Let $\mathcal{Y}$ be the label space for the classification task. Denote the forward diffusion process by $D : \mathcal{X} \to \mathcal{X}$, the reverse process by $R : \mathcal{X} \to \mathcal{X}$, and the classifier by $C : \mathcal{X} \to \mathcal{Y}$. The overall prediction function is formulated as:

$$h(\mathbf{x}) = C \circ R \circ D(\mathbf{x}), \quad \text{with } \mathbf{x} = \mathbf{x}_0. \tag{4}$$

In this context, $\mathbf{x}_T = D(\mathbf{x}_0)$ refers to the noisy image obtained after $T$ steps of diffusion, and $\hat{\mathbf{x}}_0 = R(\mathbf{x}_T)$ represents the corresponding recovered images through the reverse process. Specifically, these methods predetermine a *constant* noise level $T = t^*$ for all samples, following a shared noise schedule $\{\beta_t\}_{t \in [0,T]}$. The outcome of the forward process defined in Eq. (1) can be expressed as:

$$\mathbf{x}_T = \sqrt{\prod_{i=1}^{T}(1 - \beta_i)}\mathbf{x} + \sqrt{1 - \prod_{i=1}^{T}(1 - \beta_i)}\boldsymbol{\epsilon} \quad \text{with } \mathbf{x} = \mathbf{x}_0, \forall \mathbf{x} \in \mathcal{X},$$

where $\mathbf{x}_0$ represents the original data, and $\boldsymbol{\epsilon} \sim \mathcal{N}(\mathbf{0}, \mathbf{I})$ denotes the Gaussian noise.

**From sample-shared to sample-specific noise level**. SSNI takes a step further by transforming the sample-shared noise level $T_{\text{SH}}(\mathbf{x}) = t^*$ into a *sample-specific* noise level $T_{\text{SI}}(\mathbf{x}) = t(\mathbf{x})$, which

---

**Algorithm 1** Diffusion-based Purification with SSNI.

---

**Input:** test samples $\mathbf{x}$, a score network $s_\theta$, a reweighting function $f(\cdot)$ and a pre-determined noise level $t^*$.

1: Approximate the score by $s_\theta$: $s_\theta(\mathbf{x})$
2: Obtain the sample-specific noise level: $t(\mathbf{x}) = f(\|s_\theta(\mathbf{x})\|, t^*)$
3: The forward diffusion process: $\mathbf{x}_{t(\mathbf{x})} = \sqrt{\prod_{i=1}^{t(\mathbf{x})}(1-\beta_i)}\mathbf{x} + \sqrt{1-\prod_{i=1}^{t(\mathbf{x})}(1-\beta_i)}\boldsymbol{\epsilon}$ by Eq. (6)
4: **for** $\mathbf{t} = t(\mathbf{x})...\mathbf{1}$ **do**
5:     The reverse diffusion process: $\hat{\mathbf{x}}_{\mathbf{t}-1} = \frac{1}{\sqrt{1-\beta_{\mathbf{t}}}}\left(\hat{\mathbf{x}}_{\mathbf{t}} - \frac{\beta_{\mathbf{t}}}{\sqrt{1-\bar{\alpha}_{\mathbf{t}}}}\boldsymbol{\epsilon}_{\theta^*}(\hat{\mathbf{x}}_{\mathbf{t}}, \mathbf{t})\right) + \sqrt{\beta_{\mathbf{t}}}\boldsymbol{\epsilon}$ by Eq. (2)
6: **end for**
7: **return** purified samples $\hat{\mathbf{x}}$

---

adapts the noise injection for each sample. Given a sample-shared noise level $t^*$, SSNI defines $t(\mathbf{x})$ as

$$t(\mathbf{x}) = f(\|s_\theta(\mathbf{x})\|, t^*), \tag{5}$$

where $s_\theta(\mathbf{x})$ represents the score of a sample $\mathbf{x}$, which is estimated by the score network $s_\theta$. To avoid potential ambiguity, we have not explicitly denoted the dependency of $s_\theta(\mathbf{x})$ on $t$. However, it is important to note that the calculation of $s_\theta(\mathbf{x})$ requires a set of noise levels, which differs from the sample-shared noise level $t^*$. To further clarify, the samples $\{\mathbf{x}_i\}_{i=1}^n$ in $s_\theta(\mathbf{x}_i)$ is associated with a corresponding set of noise levels $\{t_i\}_{i=1}^n$. These individual noise levels $\{t_i\}_{i=1}^n$ are determined dynamically based on the specific characteristics of $\mathbf{x}_i$. $f(\cdot, \cdot)$ is a reweighting function that adjusts $t^*$ based on the score norm, allowing the noise level to vary depending on the sample's deviation from the clean data distribution. The outcome of the forward process of SSNI is:

$$\mathbf{x}_{t(\mathbf{x})} = \sqrt{\prod_{i=1}^{t(\mathbf{x})}(1-\beta_i)}\mathbf{x} + \sqrt{1-\prod_{i=1}^{t(\mathbf{x})}(1-\beta_i)}\boldsymbol{\epsilon} \quad \text{with } \mathbf{x} = \mathbf{x}_0, \forall \mathbf{x} \in \mathcal{X}. \tag{6}$$

In this way, SSNI enables an adaptive noise injection process tailored to the properties (i.e., score norm) of each individual sample.

## 4.2 Unifying Sample-shared and Sample-specific DBP

We define a generalized purification operator encompassing both sample-shared and sample-specific noise based DBP methods as $\Phi(\mathbf{x}) = R(\mathbf{x}_{T(\mathbf{x})})$, where $R$ denotes the reverse process, $\mathbf{x}_{T(\mathbf{x})}$ is the noisy version of $\mathbf{x}$ after $T(\mathbf{x})$ steps of diffusion, and $T : \mathcal{X} \to \mathcal{T}$ is a function that determines the noise level for each input, with $\mathcal{T} = [0, T_{\max}]$ being the range of possible noise levels. With this operator, we have the following understandings (detailed justifications are in Appendix C).

**Sample-shared DBP is a special case of SSNI.** For a sample-shared DBP, the noise level, denoted as $T_{\mathrm{SH}}(\mathbf{x}) = t^*$, is a constant for $\forall \mathbf{x} \in \mathcal{X}$, while for SSNI: $T_{\mathrm{SI}}(\mathbf{x}) = t(\mathbf{x}) = f(\|s_\theta(\mathbf{x})\|, t^*)$. Clearly, any $T_{\mathrm{SH}}(\mathbf{x})$ can be expressed by $T_{\mathrm{SI}}(\mathbf{x})$, implying that any sample-shared noise level $t^*$ is equivalently represented by SSNI with a constant reweighting function.

**SSNI has higher purification flexibility.** For a given DBP strategy, we further define the purification range $\Omega$ for an input $\mathbf{x} \in \mathcal{X}$, as the set $\Omega(\mathbf{x}) = \{\Phi(\mathbf{x}) \mid \Phi(\mathbf{x}) = R(\mathbf{x}_{\tau(\mathbf{x})}), \tau : \mathcal{X} \to \mathcal{T}\}$, which characterizes all possible purified outputs of the input $\mathbf{x}$. This concept captures the flexibility of a DBP strategy. We find that $\Omega_{\mathrm{SH}} \subseteq \Omega_{\mathrm{SI}}$ holds for any $\mathbf{x} \in \mathcal{X}$, and there exists at least one $\mathbf{x} \in \mathcal{X}$ for which the inclusion is strict, i.e., $\Omega_{\mathrm{SH}} \subsetneq \Omega_{\mathrm{SI}}$. These results show that SSNI achieves a broader purification range than sample-shared DBP, thus enabling *greater flexibility* in the purification process.

## 4.3 Realization of SSNI

In this section, we discuss the empirical realizations of SSNI in detail.

**Realization of the score.** In Section 4.1, The samples $\{\mathbf{x}_i\}_{i=1}^n$ in $s_\theta(\mathbf{x}_i)$ are associated with a corresponding set of noise levels $\{t_i\}_{i=1}^n$. However, a direct realization of this idea presents two challenges: (1) If each sample $\mathbf{x}_i$ corresponds to a single noise level $t_i$, the resulting score calculation becomes highly sensitive to the choice of $t_i$ (Zhang et al., 2023). (2) It is difficult to pre-determine

an optimal set of noise levels $\{t_i\}_{i=1}^n$ when calculating $s_\theta(\mathbf{x}_i)$, which still remains as an open question. To address the above-mentioned limitations, following Zhang et al. (2023), we use *expected perturbation score* (EPS) to measure how much a data point deviates from the clean distribution, which is defined as:

$$\text{EPS}(\mathbf{x}) = \mathbb{E}_{t \sim U(0,T')} \nabla_{\mathbf{x}} \log p_t(\mathbf{x}), \tag{7}$$

where $p_t(\mathbf{x})$ is the marginal probability density and $T'$ is the maximum noise level for EPS. EPS computes the expectation of the scores of perturbed images across different noise levels $t \sim U(0, T')$, making it more invariant to the changes in noise levels. Notably, this $T'$ is *different* from the optimal noise level $t^*$ in this paper. Following Zhang et al. (2023), we set $T' = 20$. In practice, a score $\nabla_{\mathbf{x}} \log_{p_t}(\mathbf{x})$ can be approximated by $s_\theta(\mathbf{x})$, where $s_\theta$ is a score network. In this paper, we use a pre-trained score network that has a training objective of score matching (Song & Ermon, 2019) to achieve the estimation for the score $s_\theta(\mathbf{x})$.

**Realization of the linear reweighting function.** We first design a linear function to reweight $t^*$:

$$f_{\text{linear}}(\|\text{EPS}(\mathbf{x})\|, \, t^*) = \frac{\|\text{EPS}(\mathbf{x})\| - \xi_{\min}}{\xi_{\max} - \xi_{\min}} \times t^* + b, \tag{8}$$

where $b$ is a bias term and $t^*$ denotes the optimal sample-shared noise level selected by Nie et al. (2022). Specifically, we extract 5,000 validation clean examples from the training data (denoted as $\mathbf{x}_v$) and we use $\|\text{EPS}(\mathbf{x}_v)\|$ as a *reference* to indicate the approximate EPS norm values of clean data, which can help us reweight $t^*$. Then we define:

$$\xi_{\min} = \min(\|\text{EPS}(\mathbf{x})\|, \, \|\text{EPS}(\mathbf{x}_v)\|), \, \xi_{\max} = \max(\|\text{EPS}(\mathbf{x})\|, \, \|\text{EPS}(\mathbf{x}_v)\|).$$

The key idea is to normalize $\|\text{EPS}(\mathbf{x})\|$ such that the coefficient of $t^*$ is within a range of $[0, 1]$, ensuring that the reweighted $t^*$ stays positive and avoids unbounded growth, thus preserving the semantic information.

**Realization of the non-linear reweighting function.** We then design a non-linear function based on the sigmoid function, which has two horizontal asymptotes:

$$f_{\sigma}(\|\text{EPS}(\mathbf{x})\|, \, t^*) = \frac{t^* + b}{1 + \exp\{-(\|\text{EPS}(\mathbf{x})\| - \mu)/\tau\}}, \tag{9}$$

where $b$ is a bias term and $t^*$ denotes the optimal sample-shared noise level selected by Nie et al. (2022) and $\tau$ is a temperature coefficient that controls the sharpness of the function. We denote the mean value of $\|\text{EPS}(\mathbf{x}_v)\|$ as $\mu$. This ensures that when the difference between $\|\text{EPS}(\mathbf{x})\|$ and $\mu$ is large, the reweighted $t^*$ can approach to the maximum $t^*$ in a more smooth way, and vice versa.

**Adding a bias term to the reweighting function.** One limitation of the above-mentioned reweighting functions is that *the reweighted $t^*$ cannot exceed the original $t^*$*, which may result in some adversarial noise not being removed for some adversarial examples. To address this issue, we introduce an extra bias term (i.e., $b$) to the reweighting function, which can increase the upper bound of the reweighted $t^*$ so that the maximum possible reweighted $t^*$ can exceed original $t^*$. Empirically, we find that this can further improve the robust accuracy without compromising the clean accuracy.

## 5 EXPERIMENTS

In this section, we use *SSNI-L* to denote our method with the *linear* reweighting function, and use *SSNI-N* to denote our method with the *non-linear* reweighting function.

### 5.1 EXPERIMENTAL SETTINGS

**Datasets and model architectures.** We consider two datasets for our evaluations: CIFAR-10 (Krizhevsky et al., 2009), and ImageNet-1K (Deng et al., 2009). For classificaion models, we use the pre-trained WideResNet-28-10 and WideResNet-70-16 for CIFAR-10, and the pre-trained ResNet-50 for ImageNet-1K. For diffusion models, we employed two off-the-shelf diffusion models pre-trained on CIFAR-10 and ImageNet-1K (Song et al., 2021b; Dhariwal & Nichol, 2021).

**Evaluation metrics.** For all experiments, we consider the standard accuracy (i.e., accuracy on clean examples) and robust accuracy (i.e., accuracy on adversarial examples) as the evaluation metrics.

Table 1: Standard and robust accuracy of DBP methods against adaptive white-box PGD+EOT (left: $\ell_\infty(\epsilon = 8/255)$, right: $\ell_2(\epsilon = 0.5)$) on *CIFAR-10*. WideResNet-28-10 and WideResNet-70-16 are used as classifiers. We compare the result of DBP methods with and without *SSNI-N*. We report mean and standard deviation over three runs. We show the most successful defense in **bold**. The performance improvements and degradation are reported in green and red.

| | | PGD+EOT $\ell_\infty$ ($\epsilon = 8/255$) | | | PGD+EOT $\ell_2$ ($\epsilon = 0.5$) | |
|---|---|---|---|---|---|---|
| | DBP Method | Standard | Robust | DBP Method | Standard | Robust |
| **WRN-28-10** | Nie et al. (2022) | 89.71±0.72 | 47.98±0.64 | Nie et al. (2022) | 91.80±0.84 | **82.81±0.97** |
| | + *SSNI-N* | **93.29±0.37 (+3.58)** | **48.63±0.56 (+0.65)** | + *SSNI-N* | **93.95±0.70 (+2.15)** | 82.75±1.01 (-0.06) |
| | Wang et al. (2022) | 92.45±0.64 | 36.72±1.05 | Wang et al. (2022) | 92.45±0.64 | 82.29±0.82 |
| | + *SSNI-N* | **94.08±0.33 (+1.63)** | **40.95±0.65 (+4.23)** | + *SSNI-N* | **94.08±0.33 (+1.63)** | **82.49±0.75 (+0.20)** |
| | Lee & Kim (2023) | 90.10±0.18 | 56.05±1.11 | Lee & Kim (2023) | 90.10±0.18 | 83.66±0.46 |
| | + *SSNI-N* | **93.55±0.55 (+2.66)** | **56.45±0.28 (+0.40)** | + *SSNI-N* | **93.55±0.55 (+3.45)** | **84.05±0.33 (+0.39)** |
| **WRN-70-16** | Nie et al. (2022) | 90.89±1.13 | 52.15±0.30 | Nie et al. (2022) | 92.90±0.40 | 82.94±1.13 |
| | + *SSNI-N* | **94.47±0.51 (+3.58)** | **52.47±0.66 (+0.32)** | + *SSNI-N* | **95.12±0.58 (+2.22)** | **84.38±0.58 (+1.44)** |
| | Wang et al. (2022) | 93.10±0.51 | 43.55±0.58 | Wang et al. (2022) | 93.10±0.51 | **85.03±0.49** |
| | + *SSNI-N* | **95.57±0.24 (+2.47)** | **46.03±1.33 (+2.48)** | + *SSNI-N* | **95.57±0.24 (+2.47)** | 84.64±0.51 (-0.39) |
| | Lee & Kim (2023) | 89.39±1.12 | 56.97±0.33 | Lee & Kim (2023) | 89.39±1.12 | 84.51±0.37 |
| | + *SSNI-N* | **93.82±0.24 (+4.44)** | **57.03±0.28 (+0.06)** | + *SSNI-N* | **93.82±0.24 (+4.43)** | **84.83±0.33 (+0.32)** |

**Baseline settings.** We use three well-known DBP methods as our baselines: *DiffPure* (Nie et al., 2022), *GDMP* (Wang et al., 2022) and *GNS* (Lee & Kim, 2023). The detailed configurations of baseline methods can be found in Appendix D. For the reverse process within diffusion models, we consider DDPM sampling method (Ho et al., 2020) in the DBP methods.

**Evaluation settings for DBP baselines.** Following Lee & Kim (2023), we use a fixed subset of 512 randomly sampled images for all evaluations due to high computational cost of applying adaptive white-box attacks to DBP methods. Lee & Kim (2023) provide a robust evaluation framework for existing DBP methods and demonstrate that PGD+EOT is the golden standard for DBP evaluations. Therefore, following Lee & Kim (2023), we mainly use adaptive white-box PGD+EOT attack with 200 PGD iterations for CIFAR-10 and 20 PGD iterations for ImageNet-1K. We use 20 EOT iterations for all experiments to mitigate the stochasticity introduced by the diffusion models. As PGD is a gradient-based attack, we compute the gradients of the entire process from a surrogate process (Lee & Kim, 2023). The details of the surrogate process is explained in Appendix E. We also evaluate DBP methods under adaptive BPDA+EOT attack, which leverages an identity function to approximate the direct gradient rather than direct computing the gradient of the defense system.

**Evaluation settings for SSNI.** Since SSNI introduces an extra reweighting process than DBP baselines, we implicitly design two adaptive white-box attacks by considering the *entire defense mechanism* of SSNI (i.e., adaptive white-box PGD+EOT attack and adaptive white-box BPDA+EOT attack). *To make a fair comparison, we evaluate SSNI on adaptive white-box attacks with the same configurations mentioned above*. The algorithmic descriptions for the adaptive white-box PGD+EOT attack and adaptive white-box BPDA+EOT attack is provided in Appendix F and G.

## 5.2 DEFENDING AGAINST ADAPTIVE WHITE-BOX PGD+EOT

We mainly present and analyze the evaluation results of *SSNI-N* in this section and the experimental results of *SSNI-L* can be found in Appendix H.

**Result analysis on CIFAR-10.** Table 1 shows the standard and robust accuracy against PGD+EOT $\ell_\infty(\epsilon = 8/255)$ and $\ell_2(\epsilon = 0.5)$ threat models on CIFAR-10, respectively. Notably, *SSNI-N* effectively improves the accuracy-robustness trade-off on PGD+EOT $\ell_\infty(\epsilon = 8/255)$ compared to DBP baselines. Specifically, *SSNI-N* improves standard accuracy of *DiffPure* by 3.58% on WideResNet-28-10 and WideResNet-70-16 without compromising robust accuracy. For *GDMP*, the standard accuracy grows by 1.63% on WideResNet-28-10 and by 2.47% on WideResNet-70-16, respectively. Notably, *SSNI-N* improves the robust accuracy of *GDMP* by 4.23% on WideResNet-28-10 and by 2.48% on WideResNet-70-16. For *GNS*, both the standard accuracy and robust accuracy are improved by a notable margin. We can observe a similar trend in PGD+EOT $\ell_2(\epsilon = 0.5)$. Despite some decreases in robust accuracy (e.g., 0.06% on *DiffPure* and 0.39% on *GDMP*), *SSNI-N* can improve standard accuracy by a notable margin, and thus improving accuracy-robustness trade-off.

Table 2: Standard and robust accuracy (%) against adaptive white-box PGD+EOT $\ell_\infty(\epsilon = 4/255)$ on *ImageNet-1K*.

Table 3: Standard and robust accuracy (%) against adaptive white-box BPDA+EOT $\ell_\infty(\epsilon = 8/255)$ attack on *CIFAR-10*.

| | PGD+EOT $\ell_\infty$ ($\epsilon = 4/255$) | | |
|---|---|---|---|
| | DBP Method | Standard | Robust |
| RN-50 | Nie et al. (2022) | 68.23±0.92 | 30.34±0.72 |
| | + *SSNI-N* | **70.25±0.56 (+2.02)** | **33.66±1.04 (+3.32)** |
| | Wang et al. (2022) | 74.22±0.12 | 0.39±0.03 |
| | + *SSNI-N* | **75.07±0.18 (+0.85)** | **5.21±0.24 (+4.82)** |
| | Lee & Kim (2023) | 70.18±0.60 | 42.45±0.92 |
| | + *SSNI-N* | **72.69±0.80 (+2.51)** | **43.48±0.25 (+1.03)** |

| | BPDA+EOT $\ell_\infty$ ($\epsilon = 8/255$) | | |
|---|---|---|---|
| | DBP Method | Standard | Robust |
| WRN-28-10 | Nie et al. (2022) | 89.71±0.72 | 81.90±0.49 |
| | + *SSNI-N* | **93.29±0.37 (+3.58)** | **82.10±1.15 (+0.20)** |
| | Wang et al. (2022) | 92.45±0.64 | 79.88±0.89 |
| | + *SSNI-N* | **94.08±0.33 (+1.63)** | **80.99±1.09 (+1.11)** |
| | Lee & Kim (2023) | 90.10±0.18 | **88.40±0.88** |
| | + *SSNI-N* | **93.55±0.55 (+3.45)** | 87.30±0.42 (-1.10) |

**Result analysis on ImageNet-1K.** Table 2 presents the evaluation results against adaptive white-box PGD+EOT $\ell_\infty(\epsilon = 4/255)$ on ImageNet-1K. *SSNI-N* outperforms all baseline methods by notably improving both the standard and robust accuracy, which demonstrates the effectiveness of *SSNI-N* in defending against strong white-box adaptive attack and indicates the strong scalability of SSNI on large-scale datasets such as ImageNet-1K.

## 5.3 DEFENDING AGAINST ADAPTIVE WHITE-BOX BPDA+EOT

We mainly present and analyze the evaluation results of *SSNI-N* in this section and the experimental results of *SSNI-L* can be found in Appendix H.

We further evaluate the performance of *SSNI-N* against adaptive white-box BPDA+EOT $\ell_\infty(\epsilon = 8/255)$, which is an adaptive attack specifically designed for DBP methods (Tramèr et al., 2020; Hill et al., 2021), as demonstrated in Table 6. Specifically, incorporating *SSNI-N* with *DiffPure* can further improve the standard accuracy by 3.58% without compromising robust accuracy. Notably, incorporating *SSNI-N* with *GDMP* can improve the standard and robust accuracy *simultaneously* by a large margin. Despite some decreases in robust accuracy when incorporating *SSNI-N* with *GNS* (i.e., 1.10%), *SSNI-N* can improve standard accuracy significantly (i.e., 3.45%), and thus improving the accuracy-robustness trade-off by a notable margin.

## 5.4 ABLATION STUDY

**Ablation study on $\tau$ in *SSNI-N*.** We investigate how the temperature coefficient $\tau$ in Eq. (9) affects the performance of *SSNI-N* against adaptive white-box PGD+EOT $\ell_\infty(\epsilon = 8/255)$ attack on CIFAR-10 in Figure 3. The temperature coefficient $\tau$ controls the sharpness of the curve of the *non-linear* reweighting function. A higher $\tau$ leads to a more smooth transition between the low and high values of the reweighting function, resulting in less sensitivity to the changes of the input. From Figure 3, the standard accuracy remains stable across different $\tau$s, while the robust accuracy increases to the climax when $\tau = 20$. Therefore, we choose $\tau^* = 20$ for the non-linear reweighting function to optimize the accuracy-robustness trade-off for DBP methods.

**Ablation study on sampling methods.** *DiffPure* originally used an adjoint method to efficiently compute the gradients of the system, but Lee & Kim (2023) and Chen et al. (2024) suggest to replace adjoint solver with sdeint solver for the purpose of computing full gradients more accurately (Li et al., 2020; Kidger et al., 2021). Therefore, we investigate whether using different sampling methods affect the performance of DBP methods (here we use *DiffPure* as the baseline method). We further compare the results with DDIM sampling method (Song et al., 2021a), which is a faster sampling method than DDPM (Ho et al., 2020). From Table 4, DDPM achieves the best accuracy-robustness trade-off among the three sampling methods, and thus we select DDPM as the sampling method for all baseline methods in this paper.

**Ablation study on score norms.** We investigate the effect of using single score norm (i.e., $\|\nabla_\mathbf{x} \log p_t(\mathbf{x})\|$) for SSNI in Appendix I. We find that although single score norm can notably improve the standard accuracy, it suffers from the degradation in robust accuracy. This might be attributed to the fact that single score norm is sensitive to the purification noise levels.

Table 4: Ablation study on different sampling methods during the reverse diffusion process. We measure the standard and robust accuracy (%) against PGD+EOT $\ell_\infty(\epsilon = 8/255)$ on *CIFAR-10*. We use *DiffPure* as the baseline method and we set $t^* = 100$. WideResNet-28-10 is used as the classifier. We report mean and the standard deviations over three runs.

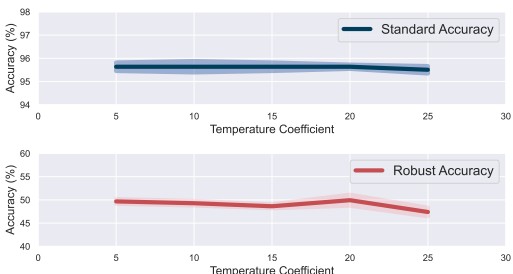

| Sampling Method | Standard | Robust |
|---|---|---|
| sdeint solver | 89.06±0.48 | 47.72±0.24 |
| DDPM | 89.71±0.72 | 47.98±0.64 |
| DDIM | 91.54±0.72 | 37.5±0.80 |

Figure 3: **Top:** standard accuracy (%) vs. $\tau$; **Bottom:** robust accuracy (%) vs. $\tau$. We report mean and the standard deviations over three runs.

**Ablation study on bias terms** $b$**.** We investigate the effect of the bias term $b$ in reweighting functions to the performance of our framework in Appendix J. We find that the selection of bias term will not significantly impact the performance of our framework under CIFAR-10 and ImageNet-1K. Note that when bias increases, there is a general observation that the clean accuracy drops and the robust accuracy increases. This perfectly aligns with the understanding of optimal noise level selections in existing DBP methods, where large noise level would lead to drop of both clean and robust accuracy and small noise level cannot remove the adversarial perturbation effectively.

### 5.5 INFERENCE TIME OF SSNI

The inference time (in seconds) for incorporating SSNI modules into existing DBP methods on CIFAR-10 and ImageNet-1K are reported in Appendix K. The inference time is measured as the time it takes for a single test image to complete the purification process. Specifically, SSNI is approximately 0.5 seconds slower than baseline methods on CIFAR-10 and 5 seconds slower than baseline methods on ImageNet-1K. Thus, compared with DBP baseline methods, this reweighting process is *lightweight*, ensuring that SSNI is computationally feasible and can be applied in practice with minimal overhead. The details of computing resources can also be found in Appendix K.

## 6 LIMITATION

**The design of reweighting functions.** The proposed reweighting functions (i.e., the linear and non-linear reweighting functions) may not be the optimal ones for SSNI. However, designing an effective reweighting function is an open question, and we leave it as future work.

**Extra computational cost.** The integration of an extra reweighting process will inevitably bring some extra cost. Luckily, we find that this reweighting process is *lightweight*, making SSNI computationally feasible compared to existing DBP methods (see Section 5.5).

## 7 CONCLUSION

In this paper, we find that an optimal $t^*$ indeed could be different at sample-level. Motivated by this finding, we propose a new framework called *Sample-specific Score-aware Noise Injection* (SSNI). SSNI sample-wisely reweight $t^*$ for each sample based on its score norm, which generally injects less noise to clean samples and sufficient noise to adversarial samples, leading to a notable improvement in accuracy-robustness trade-off. We hope this simple yet effective framework could open up a new perspective in DBP methods and lay the groundwork for sample specific noise injections.

### ETHICS STATEMENT

This study on adversarial defense mechanisms raises important ethical considerations that we have carefully addressed. We have taken steps to ensure our adversarial defense method is fair. We use widely accepted public benchmark datasets to ensure comparability of our results. Our evaluation

encompasses a wide range of attack types and strengths to provide a comprehensive assessment of our defense mechanism.

We have also carefully considered the broader impacts of our work. The proposed defense algorithm contributes to the development of more robust machine learning models, potentially improving the reliability of AI systems in various applications. We will actively engage with the research community to promote responsible development and use of adversarial defenses.

## REPRODUCIBILITY STATEMENT

Appendix A include justifications of the theoretical results in Section 3. To replicate the experimental results presented in Section 5, we have included a link to our anonymous downloadable source code in the abstract. We include additional implementation details required to reproduce the reported results in Appendix D and E.

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

## A DETAILED PROOFS

**Definition 1** (Marginal Probability Density). Let $\mathcal{X}$ be the image sample space, $p(\mathbf{x}_0)$ be the natural data distribution over $\mathcal{X}$, and the diffusion process kernel defined in Eq. (1). The marginal probability density $p_t : \mathcal{X} \to \mathbb{R}_+$ at time $0 \le t \le T$ can be expressed as:

$$p_t(\mathbf{x}) = \int_{\mathcal{X}} q(\mathbf{x}|\mathbf{x}_0)p(\mathbf{x}_0)d\mathbf{x}_0, \tag{10}$$

where $q(\mathbf{x}|\mathbf{x}_0)$ describes how a natural sample $\mathbf{x}_0$ evolves under the forward process to $\mathbf{x}$ at time $t$.

Before proceeding with the proof of Lemma 1 and Proposition 1, we start by presenting two lemmas to facilitate the proof.

**Lemma 2.** *Let $p_t(\mathbf{x})$ denote the marginal probability density of $\mathbf{x}$ at time $t$. For any $\mathbf{x} \in \mathcal{X}$ and $0 \le t \le T$, the score function $\nabla_{\mathbf{x}} \log p_t(\mathbf{x})$ at time $t$ can be expressed as:*

$$\nabla_{\mathbf{x}} \log p_t(\mathbf{x}) = \mathbb{E}_{p(\mathbf{x}_0|\mathbf{x})}\left[\nabla_{\mathbf{x}} \log q(\mathbf{x}|\mathbf{x}_0)\right],$$

*where $p(\mathbf{x}_0|\mathbf{x})$ is the posterior given by Bayes' Rule:*

$$p(\mathbf{x}_0|\mathbf{x}) = \frac{q(\mathbf{x}|\mathbf{x}_0)p(\mathbf{x}_0)}{\int q(\mathbf{x}|\mathbf{x}_0)p(\mathbf{x}_0)d\mathbf{x}_0}.$$

*Proof.*

$$
\begin{aligned}
\nabla_{\mathbf{x}} \log p_t(\mathbf{x}) &= \nabla_{\mathbf{x}} \log \int q(\mathbf{x}|\mathbf{x}_0)p(\mathbf{x}_0)d\mathbf{x}_0 && \text{(by Def. 1)} \\
&= \frac{\nabla_{\mathbf{x}} \int q(\mathbf{x}|\mathbf{x}_0)p(\mathbf{x}_0)d\mathbf{x}_0}{\int q(\mathbf{x}|\mathbf{x}_0)p(\mathbf{x}_0)d\mathbf{x}_0} && \text{(chain rule)} \\
&= \frac{\int \nabla_{\mathbf{x}} q(\mathbf{x}|\mathbf{x}_0)p(\mathbf{x}_0)d\mathbf{x}_0}{\int q(\mathbf{x}|\mathbf{x}_0)p(\mathbf{x}_0)d\mathbf{x}_0} && \text{(Leibniz integral rule)} \\
&= \frac{\int \frac{\nabla_{\mathbf{x}} q(\mathbf{x}|\mathbf{x}_0)}{q(\mathbf{x}|\mathbf{x}_0)} q(\mathbf{x}|\mathbf{x}_0)p(\mathbf{x}_0)d\mathbf{x}_0}{\int q(\mathbf{x}|\mathbf{x}_0)p(\mathbf{x}_0)d\mathbf{x}_0} && \text{(manipulate } q(\mathbf{x}|\mathbf{x}_0)) \\
&= \frac{\int (\nabla_{\mathbf{x}} \log q(\mathbf{x}|\mathbf{x}_0)) q(\mathbf{x}|\mathbf{x}_0)p(\mathbf{x}_0)d\mathbf{x}_0}{\int q(\mathbf{x}|\mathbf{x}_0)p(\mathbf{x}_0)d\mathbf{x}_0} && \text{(chain rule)} \\
&= \int (\nabla_{\mathbf{x}} \log q(\mathbf{x}|\mathbf{x}_0)) p(\mathbf{x}_0|\mathbf{x})d\mathbf{x}_0 && (p(\mathbf{x}_0|\mathbf{x}) \triangleq \frac{q(\mathbf{x}|\mathbf{x}_0)p(\mathbf{x}_0)}{\int q(\mathbf{x}|\mathbf{x}_0)p(\mathbf{x}_0)d\mathbf{x}_0})
\end{aligned} \tag{11}
$$

This ends up with the expectation over the posterior $p(\mathbf{x}_0|\mathbf{x})$ as $\mathbb{E}_{p(\mathbf{x}_0|\mathbf{x})}\left[\nabla_{\mathbf{x}} \log q(\mathbf{x}|\mathbf{x}_0)\right]$. $\qquad\square$

Lemma 2 establishes a relationship between the marginal density and the forward process, where we can express the behavior of $\nabla_{\mathbf{x}} \log p_t(\mathbf{x})$ in terms of the well-defined $q(\mathbf{x}|\mathbf{x}_0)$.

**Lemma 3.** *Consider the forward process $q(\mathbf{x}_t|\mathbf{x}_0)$ is defined with noise schedule $\{\beta_t\}_{t\in[0,T]}$, where $\beta_t \in (0,1)$ for all $0 \le t \le T$. Then, for any $\mathbf{x} \in \mathcal{X}$ and $0 \le t \le T$, the score function $\nabla_{\mathbf{x}} \log p_t(\mathbf{x})$ can be decomposed as:*

$$\nabla_{\mathbf{x}} \log p_t(\mathbf{x}) = -g(t)\mathbf{x} + \mathbb{E}_{p(\mathbf{x}_0|\mathbf{x})}\left[g(t)\sqrt{\bar{\alpha}_t}\mathbf{x}_0\right],$$

*where $g(t) = 1/(1 - \bar{\alpha}_t)$ and $\bar{\alpha}_t = \prod_{i=1}^{t}(1 - \beta_i)$.*

*Proof.* Denote the score function $s_t(\mathbf{x}) = \nabla_{\mathbf{x}} \log p_t(\mathbf{x})$ for convenience. By applying the results from Lemma 2, we write:

$$s_t(\mathbf{x}) = \mathbb{E}_{p(\mathbf{x}_0|\mathbf{x})}\left[q(\mathbf{x}|\mathbf{x}_0)\right].$$

Provided that $q(\mathbf{x}|\mathbf{x}_0) = \mathcal{N}\left(\mathbf{x}; \sqrt{\bar{\alpha}_t}\mathbf{x}_0, (1 - \bar{\alpha}_t)\mathbf{I}\right)$, we compute the gradient of the log-probability of the forward process as:

$$\nabla_{\mathbf{x}} \log q(\mathbf{x}|\mathbf{x}_0) = \nabla_{\mathbf{x}} \mathcal{N}\left(\mathbf{x}; \sqrt{\bar{\alpha}_t}\mathbf{x}_0, (1 - \bar{\alpha}_t)\mathbf{I}\right)$$

$$= \nabla_{\mathbf{x}} \left[ -\frac{1}{2(1 - \bar{\alpha}_t)}(\mathbf{x} - \sqrt{\bar{\alpha}_t}\mathbf{x}_0)^{\top}(\mathbf{x} - \sqrt{\bar{\alpha}_t}\mathbf{x}_0) + C \right]$$

$$= -\frac{1}{1 - \bar{\alpha}_t}(\mathbf{x} - \sqrt{\bar{\alpha}_t}\mathbf{x}_0),$$

where $C$ is a constant term independent of $\mathbf{x}$. Substitute the result back into $s_t(\mathbf{x})$, we have:

$$s_t(\mathbf{x}) = \mathbb{E}_{p(\mathbf{x}_0|\mathbf{x})}\left[q(\mathbf{x}|\mathbf{x}_0)\right]$$

$$= -\frac{1}{1 - \bar{\alpha}_t}\mathbb{E}_{p(\mathbf{x}_0|\mathbf{x})}\left[\mathbf{x} - \sqrt{\bar{\alpha}_t}\mathbf{x}_0\right]$$

$$= -\frac{1}{1 - \bar{\alpha}_t}\mathbf{x} + \frac{\sqrt{\bar{\alpha}_t}}{1 - \bar{\alpha}_t}\mathbb{E}_{p(\mathbf{x}_0|\mathbf{x})}[\mathbf{x}_0]$$

$$= -g(t)\mathbf{x} + \mathbb{E}_{p(\mathbf{x}_0|\mathbf{x})}\left[g(t)\sqrt{\bar{\alpha}_t}\mathbf{x}_0\right].$$

The Lemma is completed by substituting $g(t) = \frac{1}{1-\bar{\alpha}_t}$ into the last equation. $\qquad\square$

Lemma 3 expresses the score function by the noisy data $\mathbf{x}$ with noise level $t$ and the clean data $\mathbf{x}_0$. We next examine how the score norm evolves to further understand its behavior as time increases.

**Lemma 4** (*cf.* Lemma 1). *Suppose there exists a constant $K > 0$ such that for all $t \geq 0$ and all $\mathbf{x}_t \in \mathcal{X}$, the expected norm of the clean data given $\mathbf{x}_t$ satisfies: $\mathbb{E}_{\mathbf{x}_0 \sim p(\mathbf{x}_0|\mathbf{x})}[\|\mathbf{x}_0\|] \leq K\|\mathbf{x}_t\|$. Then, there exist constants $0 < C < 1$ and $T_0 > 0$ such that for all $t \geq T_0$:*

$$\frac{\|\nabla_{\mathbf{x}} \log p_t(\mathbf{x})\|}{\|\mathbf{x}\|} > C.$$

*Proof.* Again, denote $s_t(\mathbf{x}) = \nabla_{\mathbf{x}} \log p_t(\mathbf{x})$. From Lemma 3, we have:

$$s_t(\mathbf{x}) = -g(t)\mathbf{x} + \mathbb{E}_{p(\mathbf{x}_0|\mathbf{x})}\left[g(t)\sqrt{\bar{\alpha}_t}\mathbf{x}_0\right].$$

Applying the triangle inequality to it, we have:

$$\|s_t(\mathbf{x})\| \geq g(t)\|\mathbf{x}\| - \left\|\mathbb{E}_{p(\mathbf{x}_0|\mathbf{x})}\left[g(t)\sqrt{\bar{\alpha}_t}\mathbf{x}_0\right]\right\|$$

$$\geq g(t)\|\mathbf{x}\| - g(t)\sqrt{\bar{\alpha}_t}\mathbb{E}_{p(\mathbf{x}_0|\mathbf{x})}\left[\|\mathbf{x}_0\|\right],$$

where the second inequality comes from applying Jensen's inequality: $f(\mathbb{E}[X]) \leq \mathbb{E}[f(X)]$ for convex function $f$ and random variable $X$, which leads to:

$$\left\|\mathbb{E}_{p(\mathbf{x}_0|\mathbf{x})}\left[g(t)\sqrt{\bar{\alpha}_t}\mathbf{x}_0\right]\right\| \leq \mathbb{E}_{p(\mathbf{x}_0|\mathbf{x})}\left[\left\|g(t)\sqrt{\bar{\alpha}_t}\mathbf{x}_0\right\|\right]$$

$$= g(t)\sqrt{\bar{\alpha}_t}\mathbb{E}_{p(\mathbf{x}_0|\mathbf{x})}\left[\|\mathbf{x}_0\|\right].$$

As we have assumed the existence of $K > 0$ that makes $\mathbb{E}_{\mathbf{x}_0 \sim p(\mathbf{x}_0|\mathbf{x})}[\|\mathbf{x}_0\|] \leq K\|\mathbf{x}_t\|$ holds for all $\mathbf{x} \in \mathcal{X}$, we have:

$$\|s_t(\mathbf{x})\| \geq g(t)\|\mathbf{x}\| - g(t)\sqrt{\bar{\alpha}_t}K\|\mathbf{x}\|$$

$$= g(t)\|\mathbf{x}\|(1 - K\sqrt{\bar{\alpha}_t}).$$

We then turn to look into the asymptotic behavior of $g(t)$. We have $\bar{\alpha}_t = \prod_{s=1}^{t}(1 - \beta_s)$. Since $0 < 1 - \beta_s < 1$ for all $s$, and the sequence is decreasing (as $\beta_s$ is increasing), it is easy to check that

$$\lim_{t \to \infty} \bar{\alpha}_t = 0$$

and

$$\lim_{t \to \infty} g(t) = \lim_{t \to \infty} \frac{1}{1 - \bar{\alpha}_t} = 1.$$

This can then be formalized as, for any $\epsilon > 0$, there exists a $T_\epsilon$ such that for all $t > T_\epsilon$:

$$|g(t) - 1| < \epsilon, \text{ and } \sqrt{\bar{\alpha}_t} < \epsilon.$$

Then, for $t > T_\epsilon$, we have

$$
\begin{aligned}
\|s_t(\mathbf{x})\| &\geq g(t) \|\mathbf{x}\| (1 - K\sqrt{\bar{\alpha}_t}) \\
&> (1 - \epsilon) \|\mathbf{x}\| (1 - K\epsilon), \\
&= (1 - \epsilon - K\epsilon + K\epsilon^2) \|\mathbf{x}\| \\
&= (1 - (K+1)\epsilon + K\epsilon^2) \|\mathbf{x}\|.
\end{aligned}
$$

To establish the desired inequality $\|s_t(\mathbf{x})\| \geq C \|\mathbf{x}\|$ for constants $C > 0$, we next investigate whether this quatratic inequality $K\epsilon^2 + (K+1)\epsilon + (1-C) > 0$ can be solved. Denote the discriminant $D = (K+1)^2 - 4K(1-C)$, we have

$$
\begin{aligned}
D &= (K+1)^2 - 4K(1-C) \\
&= K^2 - 2K + 1 + 4KC \\
&= (K-1)^2 + 4KC \\
&> 0 \qquad \text{(since } (K-1)^2 \geq 0 \text{ and } K > 0).
\end{aligned}
$$

Then, let $\epsilon_1 < \epsilon_2$ be two real roots of $K\epsilon^2 + (K+1)\epsilon + (1-C) = 0$, as the parabola opens upwards, i.e., $K > 0$, note that $\epsilon > 0$, we further need to ensure the smaller root $\epsilon = \frac{K+1-\sqrt{D}}{2K} > 0$, such that

$$
\begin{aligned}
K + 1 &> \sqrt{(K+1)^2 - 4K(1-C)} \\
(K+1)^2 &> (K-1)^2 + 4KC \\
4K &> 4KC \\
1 &> C.
\end{aligned}
$$

Putting them together, we have $\epsilon \in (0, \epsilon_1) \cup (\epsilon_2, \infty)$ that makes $\|s_t(\mathbf{x})\| \geq C \|\mathbf{x}\|$ hold, given a constant $0 < C < 1$ in relation to $K$ and $\epsilon$. Setting $T_0 = T_\epsilon$ completes the proof. $\qquad\square$

**Proposition 2** (*cf.* Proposition 1). *Consider the diffusion model satisfying all conditions as specified in Lemma 1. Assume that there exist constants $K > 0$, such that $\beta_t \leq K$ for all $t \geq 0$. Additionally, suppose $\|\mathbf{x}\| \leq M$ for any $\mathbf{x} \in \mathcal{X}$, for some $M > 0$. Then, for any $\epsilon$, there exists a constant $\Delta = 2\epsilon/(CK)$ such that for $t_1, t_2 \geq 0$, we have:*

$$
\left| \|\nabla_\mathbf{x} \log p_{t_1}(\mathbf{x})\| - \|\nabla_\mathbf{x} \log p_{t_2}(\mathbf{x})\| \right| > \epsilon, \quad \text{with } |t_1 - t_2| \geq \Delta.
$$

*Proof.* Denote $s_t(\mathbf{x}) = \nabla_\mathbf{x} \log p_t(\mathbf{x})$. Recall that we can express the score function as

$$
\begin{aligned}
s_t(\mathbf{x}) &= \mathbb{E}_{p(\mathbf{x}_0|\mathbf{x})} [q(\mathbf{x}|\mathbf{x}_0)] \\
&= -\frac{1}{1-\bar{\alpha}_t} \mathbb{E}_{p(\mathbf{x}_0|\mathbf{x})} \left[ \mathbf{x} - \sqrt{\bar{\alpha}_t}\mathbf{x}_0 \right] \\
&= -\frac{1}{1-\bar{\alpha}_t} \mathbf{x} + \frac{\sqrt{\bar{\alpha}_t}}{1-\bar{\alpha}_t} \mathbb{E}_{p(\mathbf{x}_0|\mathbf{x})}[\mathbf{x}_0]
\end{aligned}
$$

Let $g(t) = \|s_t(\mathbf{x})\|^2$. We compute $\frac{\partial g(t)}{\partial t}$ using the chain rule and the product rule:

$$
\begin{aligned}
\frac{\partial g(t)}{\partial t} &= 2 \left\langle s_t(\mathbf{x}), \frac{\partial s_t(\mathbf{x})}{\partial t} \right\rangle \\
&= 2 \left\langle s_t(\mathbf{x}), \frac{\partial}{\partial t} \left( -\frac{1}{1-\bar{\alpha}_t} \mathbf{x} + \frac{\sqrt{\bar{\alpha}_t}}{1-\bar{\alpha}_t} \mathbb{E}_{p(\mathbf{x}_0|\mathbf{x})}[\mathbf{x}_0] \right) \right\rangle
\end{aligned}
$$

Recall that $\bar{\alpha}_t = \prod_{i=1}^t (1 - \beta_i)$, we next derive the derivative of $\bar{\alpha}_t$. We consider the continuous approximation of $t$, where the product is approximated as an exponential of the integral

$$
\bar{\alpha}_t = \exp\left( \int_0^t \log(1 - \beta_i) \mathrm{d}i \right).
$$

As $\beta_t$ is typically assumed to be small (which is an implicit common practice (Ho et al., 2020)), we further simplify $\log(1 - \beta_i)$ with its first-order Taylor approximation, i.e., $\log(1 - \beta_i) \approx -\beta_i$, which

thus leads to the approximated $\bar{a}_t \approx \exp\left(-\int_0^t \beta_i \mathrm{d}i\right)$. This way we compute the derivative of $\bar{a}_t$ as

$$\frac{\partial}{\partial t}\bar{a}_t = \frac{\partial}{\partial t}\exp\left(-\int_0^t \beta_i \mathrm{d}i\right)$$

$$= \exp\left(-\int_0^t \beta_i \mathrm{d}i\right)\frac{\partial}{\partial t}\left(-\int_0^t \beta_i \mathrm{d}i\right)$$

$$= \exp\left(-\int_0^t \beta_i \mathrm{d}i\right)(-\beta_t)$$

$$= -\beta_t\bar{\alpha}_t$$

We can then compute the derivatives of the coefficients:

$$\frac{\partial}{\partial t}\left(\frac{1}{1-\bar{\alpha}_t}\right) = \frac{-1}{(1-\bar{a}_t)^2}\frac{\partial}{\partial t}(1-\bar{a}_t)$$

$$= \frac{\beta_t\bar{\alpha}_t}{(1-\bar{\alpha}_t)^2}$$

$$\frac{\partial}{\partial t}\left(\frac{\sqrt{\bar{\alpha}_t}}{1-\bar{\alpha}_t}\right) = \frac{\frac{\partial}{\partial t}\sqrt{\bar{\alpha}_t}(1-\bar{\alpha}_t)-\sqrt{\bar{\alpha}_t}\frac{\partial}{\partial t}(1-\bar{\alpha}_t)}{(1-\bar{\alpha}_t)^2}$$

$$= \frac{\left(\frac{1}{2}\bar{\alpha}_t^{-1/2}\frac{\partial\bar{\alpha}_t}{\partial t}\right)(1-\bar{\alpha}_t)+\sqrt{\bar{\alpha}_t}\frac{\partial\bar{\alpha}_t}{\partial t}}{(1-\bar{\alpha}_t)^2}$$

$$= \frac{\left(\frac{1}{2}\bar{\alpha}_t^{-1/2}(-\beta_t\bar{\alpha}_t)\right)(1-\bar{\alpha}_t)+\sqrt{\bar{\alpha}_t}(-\beta_t\bar{\alpha}_t)}{(1-\bar{\alpha}_t)^2}$$

$$= \frac{-\frac{1}{2}\beta_t\bar{\alpha}_t^{1/2}(1-\bar{\alpha}_t)-\beta_t\bar{\alpha}_t^{3/2}}{(1-\bar{\alpha}_t)^2}$$

$$= -\beta_t\bar{\alpha}_t^{1/2}\frac{\frac{1}{2}(1-\bar{\alpha}_t)+\bar{\alpha}_t}{(1-\bar{\alpha}_t)^2}$$

$$= -\beta_t\sqrt{\bar{\alpha}_t}\frac{1+\bar{\alpha}_t}{2(1-\bar{\alpha}_t)^2}$$

Using the computed derivatives:

$$\frac{\partial s_t(\mathbf{x})}{\partial t} = \frac{\beta_t\bar{\alpha}_t}{(1-\bar{\alpha}_t)^2}\mathbf{x}-\beta_t\sqrt{\bar{\alpha}_t}\frac{1+\bar{\alpha}_t}{2(1-\bar{\alpha}_t)^2}\mathbb{E}_{p(\mathbf{x}_0|\mathbf{x})}[\mathbf{x}_0].$$

Substituting these back, we get:

$$\frac{\partial g}{\partial t} = 2\left\langle -\frac{1}{1-\bar{\alpha}_t}\mathbf{x}+\frac{\sqrt{\bar{\alpha}_t}}{1-\bar{\alpha}_t}\mathbb{E}_{p(\mathbf{x}_0|\mathbf{x})}[\mathbf{x}_0], \frac{\beta_t\bar{\alpha}_t}{(1-\bar{\alpha}_t)^2}\mathbf{x}-\beta_t\sqrt{\bar{\alpha}_t}\frac{1+\bar{\alpha}_t}{2(1-\bar{\alpha}_t)^2}\mathbb{E}_{p(\mathbf{x}_0|\mathbf{x})}[\mathbf{x}_0]\right\rangle$$

$$= \frac{2\beta_t\bar{\alpha}_t}{(1-\bar{\alpha}_t)^2}\left\langle s_t(\mathbf{x}), \mathbf{x}-\frac{1+\bar{\alpha}_t}{2\sqrt{\bar{\alpha}_t}}\mathbb{E}_{p(\mathbf{x}_0|\mathbf{x})}[\mathbf{x}_0]\right\rangle.$$

Next, under Cauchy-Schwarz inequality: $|\langle \mathbf{a}, \mathbf{b}\rangle| \le \|\mathbf{a}\|\|\mathbf{b}\|$, we have

$$|\frac{\partial g}{\partial t}| \le \frac{2\beta_t\bar{\alpha}_t}{(1-\bar{\alpha}_t)^2}\|s_t(\mathbf{x})\|\cdot\left\|\mathbf{x}-\frac{1+\bar{\alpha}_t}{2\sqrt{\bar{\alpha}_t}}\mathbb{E}_{p(\mathbf{x}_0|\mathbf{x})}[\mathbf{x}_0]\right\|$$

Then, by the triangle inequality for vectors $\mathbf{a}$ and $\mathbf{b}$:

$$\|\mathbf{a}-\mathbf{b}\| = \|\mathbf{a}+(-\mathbf{b})\| \le \|\mathbf{a}\|+\|-\mathbf{b}\| = \|\mathbf{a}\|+\|\mathbf{b}\|,$$

and the assumption $\|\mathbf{x}\| \le M$, we know that $\left\|\mathbb{E}_{p(\mathbf{x}_0|\mathbf{x})}[\mathbf{x}_0]\right\| \le M$ almost surely. Thus,

$$|\frac{\partial g}{\partial t}| \le \frac{2\beta_t\bar{\alpha}_t}{(1-\bar{\alpha}_t)^2}\|s_t(\mathbf{x})\|\cdot\left\|\mathbf{x}-\frac{1+\bar{\alpha}_t}{2\sqrt{\bar{\alpha}_t}}\mathbb{E}_{p(\mathbf{x}_0|\mathbf{x})}[\mathbf{x}_0]\right\|$$

$$\le \frac{2\beta_t\bar{\alpha}_t}{(1-\bar{\alpha}_t)^2}\|s_t(\mathbf{x})\|\cdot\left(\|\mathbf{x}\|+\left\|\frac{1+\bar{\alpha}_t}{2\sqrt{\bar{\alpha}_t}}\mathbb{E}_{p(\mathbf{x}_0|\mathbf{x})}[\mathbf{x}_0]\right\|\right)$$

$$\le \frac{2\beta_t\bar{\alpha}_t}{(1-\bar{\alpha}_t)^2}\|s_t(\mathbf{x})\|\left(M+\frac{1+\bar{\alpha}_t}{2\sqrt{\bar{\alpha}_t}}M\right).$$

Let $C_1 = \sup_{t \geq 0} \frac{2\bar{\alpha}_t}{(1-\bar{\alpha}_t)^2}$ and $C_2 = \sup_{t \geq 0} \frac{1+\bar{\alpha}_t}{2\sqrt{\bar{\alpha}_t}}$. As $0 < \bar{\alpha}_t < 1$, it is easy to check the maximum value of $\frac{1+\bar{\alpha}_t}{2\sqrt{\bar{\alpha}_t}}$, i.e., $C_2$ is achieved when $\bar{\alpha}_t$ gets close to 1. This also aligns with the condition where $\sup \frac{2\bar{\alpha}_t}{(1-\bar{\alpha}_t)^2}$ is reached. Thus, $\left(1 + \frac{1+\bar{\alpha}_t}{2\sqrt{\bar{\alpha}_t}}\right) M \leq (1 + C_2)M = C_3 M$ for some constants $C_3 < 2$, since $C_2 \to 1$ when $\bar{\alpha}_t \to 1$. As a result, we have a constant $C > 0$ leading to the upper bound as

$$|\frac{\partial g}{\partial t}| < C\beta_t \|s_t(\mathbf{x})\|, \quad \text{with } C = C_1 \cdot (C_3 M).$$

Mean Value Theorem states that: for any $t_1, t_2$, there exists a $\xi$ between $t_1$ and $t_2$ such that: $|g(t_2) - g(t_1)| = |\frac{\partial g}{\partial t}(\xi)||t_2 - t_1| \leq C\beta_\xi \|s_\xi(\mathbf{x})\||t_2 - t_1|$.

Applying this to $g(t)$, for any $t_1, t_2$, there exists a $\xi$ between $t_1$ and $t_2$, such that:

$$\begin{aligned}
|g(t_2) - g(t_1)| &= |\frac{\partial g}{\partial t}(\xi)||t_2 - t_1| \\
&\leq C\beta_\xi \|s_\xi(\mathbf{x})\| |t_2 - t_1| \\
&\leq CK \|s_\xi(\mathbf{x})\| |t_2 - t_1| \qquad \text{(by assumption } \beta_t \leq K)
\end{aligned}$$

Then, we can express

$$\begin{aligned}
|\|s_{t_2}(\mathbf{x})\| - \|s_{t_1}(\mathbf{x})\|| &= \frac{|\|s_{t_2}(\mathbf{x})\|^2 - \|s_{t_1}(\mathbf{x})\|^2|}{(\|s_{t_2}(\mathbf{x})\| + \|s_{t_1}(\mathbf{x})\|)} \\
&\geq \frac{CK \|s_\xi(\mathbf{x})\| |t_2 - t_1|}{\|s_{t_2}(\mathbf{x})\| + \|s_{t_1}(\mathbf{x})\|}
\end{aligned}$$

From Lemma 1, we have: there exists $C' > 0$ and $T_0 > 0$ such that for all $t \geq T_0$: $\|s_t(\mathbf{x})\| > C'\|\mathbf{x}\|$. Applying this to $t_1, t_2$, and $\xi$ with $\delta > 0$, we find that there exists a $C'$ such that:

$$\begin{aligned}
\|s_{t_1}(\mathbf{x})\| &> C' \|\mathbf{x}\| \\
\|s_{t_2}(\mathbf{x})\| &> C' \|\mathbf{x}\| \\
\|s_\xi(\mathbf{x})\| &> C' \|\mathbf{x}\|
\end{aligned}$$

Substituting these back, we have:

$$\begin{aligned}
|\|s_{t_2}(\mathbf{x})\| - \|s_{t_1}(\mathbf{x})\|| &\geq \frac{CK_2 C' \|\mathbf{x}\| |t_2 - t_1|}{2C' \|\mathbf{x}\|} \\
&= \frac{CK_2}{2}|t_2 - t_1|.
\end{aligned}$$

For any $\epsilon > 0$, let $\Delta = \frac{2\epsilon}{CK_2}$. Then for $|t_2 - t_1| \geq \Delta$, we have: $|\|s_{t_2}(\mathbf{x})\| - \|s_{t_1}(\mathbf{x})\|| > \epsilon$. This completes the proof. $\square$

## B   RELATIONSHIP BETWEEN SCORE NORMS AND PERTURBATION BUDGETS

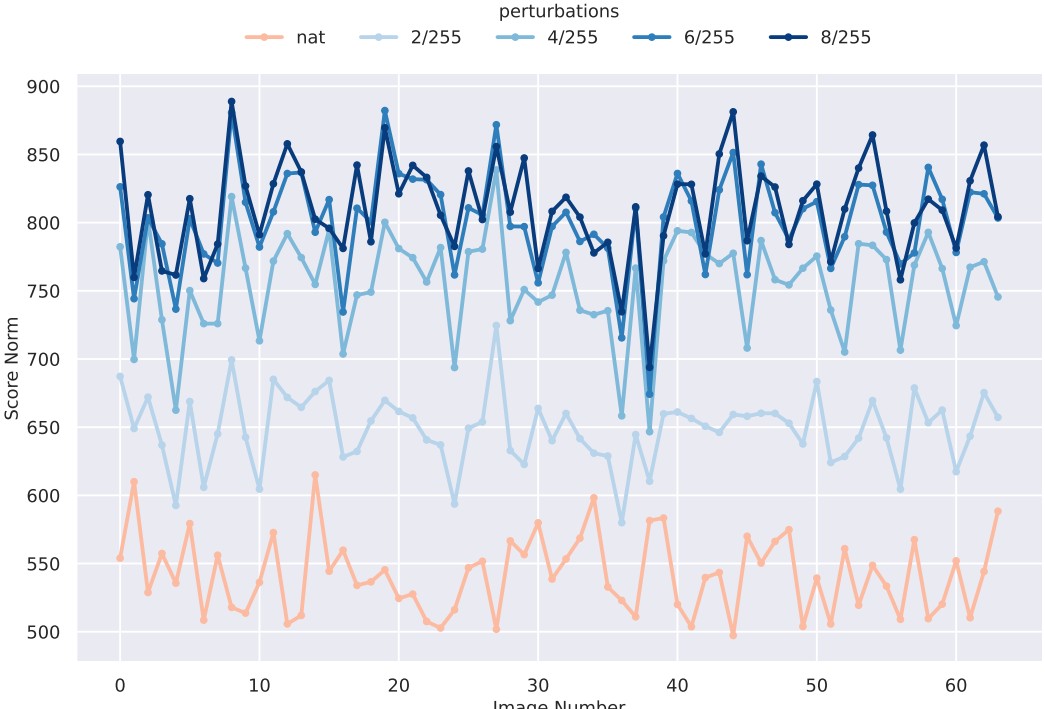

Figure 4: Relationship between score norms and perturbation budgets. We use one batch of clean data from *CIFAR-10* and employ PGD+EOT $\ell_\infty(\epsilon = 8/255)$ as the attack.

In Figure 4, we use an orange line to represent the score norm of clean data, while the score norms of adversarial data with varying perturbation budgets are shown using a gradient of blue lines. Specifically, we compute the score norm of different samples undergoing PGD+EOT $\ell_\infty(\epsilon = 8/255)$ with perturbation budgets varying between 0 and $8/255$ on CIFAR-10. The results consistently reveal that *the cleaner (i.e., lower $\epsilon$) a sample is, the smaller its score norm tends to be, and vise versa.* This further justifies the rationale of using score norms for reweighting $t^*$.

## C   JUSTIFICATION OF SECTION 4.2

**Sampled-shared DBP is a special case of SSNI**.

*Proof.* Let $\Phi_{\mathrm{SH}}$ be a sample-shared purification operator with constant noise level $t^*$. We can express $\Phi_{\mathrm{SS}}$ as a sample-specific purification operator $\Phi_{\mathrm{SI}}$ by defining the reweighting function $f$ as

$$f(z, t^*) = t^* \quad \forall z \in \mathbb{R}, t \in [0, T_{\max}].$$

Then, for any $\mathbf{x} \in \mathcal{X}$, we have

$$\begin{aligned}
\Phi_{\mathrm{SI}}(\mathbf{x}) &= R(\mathbf{x}_{T(\mathbf{x})}) \\
&= R(\mathbf{x}_{f(||s_\theta(\mathbf{x}, t^*)||, t^*)}) \\
&= R(\mathbf{x}_{t^*}) \\
&= \Phi_{\mathrm{SH}}(\mathbf{x}).
\end{aligned}$$

$\square$

**SSNI has a Higher Purification Capacity**.

Statement 1 [Comparison of Purification Range]: For any $\mathbf{x} \in \mathcal{X}$, we have $\Omega_{\mathrm{SH}}(\mathbf{x}) \subseteq \Omega_{\mathrm{SI}}(\mathbf{x})$.

*Proof.* Let $\mathbf{y} \in \Omega_{\mathrm{SH}}(\mathbf{x})$. Then $\exists t^* \in [0, T_{\max}]$ such that $\mathbf{y} = R(\mathbf{x}_{t^*})$. Define $f(z, t^*) = t^*$ for all $z$ and $t^*$, then $t(\mathbf{x}) = f(||s_\theta(\mathbf{x}, t^*)||, t^*) = t^*$. Therefore, $\mathbf{y} = R(\mathbf{x}_{t^*}) = R(\mathbf{x}_{t(\mathbf{x})}) \in \Phi_{\mathrm{SI}}(\mathbf{x})$. This completes the proof of $\Omega_{\mathrm{SH}}(\mathbf{x}) \subseteq \Omega_{\mathrm{SI}}(\mathbf{x})$. $\qquad\square$

Statement 2 [Strict Inclusion]: There exists $\mathbf{x} \in \mathcal{X}$, we have $\Omega_{\mathrm{SH}}(\mathbf{x}) \subsetneq \Omega_{\mathrm{SI}}(\mathbf{x})$.

*Proof.* Consider a non-constant score function $s_\theta(\mathbf{x}, t)$ and a non-trivial reweighting function $f$. We can choose $\mathbf{x}$ such that $t(\mathbf{x}) \neq t^*$ for any fixed $t^*$. Then $R(\mathbf{x}_{t(\mathbf{x})}) \in \Phi_{\mathrm{SI}}(\mathbf{x})$ but $R(\mathbf{x}_{t(\mathbf{x})}) \neq \Phi_{\mathrm{SH}}$. This completes the proof of $\Omega_{\mathrm{SH}}(\mathbf{x}) \subsetneq \Omega_{\mathrm{SI}}(\mathbf{x})$. $\qquad\square$

# D DEFENSE METHODS CONFIGURATIONS

For all chosen DBP methods, we utilize surrogate process to obtain gradients of the defense system during white-box adaptive attack, but we directly compute the full gradients during defense evaluation. Furthermore, we consistently apply DDPM sampling method to the selected DBP methods, which means we replace the numeric SDE solver (**sdeint**) with DDPM sampling method in Diff-Pure (Nie et al., 2022) and GDMP (Wang et al., 2022). The reason is that the SDE solver does not support sample-specific timestep input. For DDPM sampling, we can easily manipulate sample-specific timestep input by using matrix operation.

## D.1 DIFFPURE

Existing DBP methods generally follow the algorithm of DiffPure (Nie et al., 2022). DiffPure conducts evaluation on AutoAttack (Croce & Hein, 2020) and BPDA+EOT adaptive attack (Athalye et al., 2018) to measure model robustness. DiffPure chooses optimal $t^* = 100$ and $t^* = 75$ on CIFAR-10 against threat models $\ell_\infty(\epsilon = 8/255)$ and $\ell_2(\epsilon = 0.5)$, respectively. It also tests on high-resolution dataset like ImageNet-1K with $t^* = 150$ against threat models $\ell_\infty(\epsilon = 4/255)$. Calculating exact full gradients of the defense system of DiffPure is impossible since one attack iteration requires 100 function calls (with $t^* = 100$ and a step size of 1). DiffPure originally uses numerical SDE solver for calculating gradients. However, the adjoint method is insufficient to measure the model robustness since it relies on the performance of the underlying SDE solver (Zhuang et al., 2020; Lee & Kim, 2023; Chen et al., 2024). Therefore, we apply surrogate process to efficiently compute gradients of direct back-propagation in our evaluation. To overcome memory constraint issue, we align the step size settings of denoising process in adversarial attack to 5 with the evaluation settings in (Lee & Kim, 2023) and keep the timestep $t^*$ consistent with DiffPure. For ImageNet-1K evaluation, we can only afford a maximum of 10 function calls for one attack iteration.

## D.2 GDMP

GDMP basically follows the purification algorithm proposed in DiffPure (Nie et al., 2022; Wang et al., 2022), but their method further introduces a novel guidance and use multiple purification steps sequentially. GDMP proposes to use gradients of a distance between an initial input and a target being processed to preserve semantic information, shown in Eq. (12).

$$\mathbf{x}_{t-1} \sim \mathcal{N}(\boldsymbol{\mu}_\theta - s\boldsymbol{\Sigma}_\theta \nabla_{\mathbf{x}^t} \mathcal{D}(\mathbf{x}^t, \mathbf{x}_{\mathrm{adv}}^t), \boldsymbol{\Sigma}_\theta), \qquad (12)$$

Given a DDPM $(\mu_\theta(\mathbf{x}_t), \Sigma_\theta(\mathbf{x}_t))$, a gradient scale of guidance $s$. $\mathbf{x}_t$ is the data being purified, and $\mathbf{x}_{\mathrm{adv}}^t$ is the adversarial example at $t$. Also, GDMP empirically finds that multiple purification steps can improve the robustness. In the original evaluation of GDMP, the defense against the PGD attack consists of four purification steps, with each step including 36 forward steps and 36 denoising steps. For BPDA+EOT adaptive attack, GDMP uses two purification steps, each consisting of 50 forward steps and 50 denoising steps.

Lee & Kim (2023) evaluated GDMP with three types of guidance and concluded that No-Guidance provides the best robustness performance when using the surrogate process to compute the full gradient through direct backpropagation. In our evaluation, we incorporate the surrogate process with No-Guidance to evaluate GDMP. Since it is impossible to calculate the gradients of the full defense system, we use a surrogate process consisting of same number of purification steps but with larger step size in the attack (with 6 denoising steps and 10 denoising steps for PGD+EOT and

BPDA+EOT attack, respectively). Notably, GDMP only uses one purification run with 45 forward steps to evaluate on ImageNet-1K, which we keep consistent with this setting.

### D.3 GNS

Lee & Kim (2023) emphasizes the importance of selecting optimal hyperparameters in DBP methods for achieving better robust performance. Hence, Lee & Kim (2023) proposed Gradual Noise-Scheduling (GNS) for multi-step purification, which is based on the idea of choosing the best hyperparameters for multiple purification steps. It is basically the same architecture as GDMP (no guidance), but with different purification steps, forward steps and denoising steps. Specifically, GNS sets different forward and reverse diffusion steps and gradually increases the noise level at each subsequent purification step. We just keep the same hyperparameter settings and also use an ensemble of 10 runs to evaluate the method.

## E SURROGATE PROCESS OF GRADIENT COMPUTATION

The surrogate process is an efficient approach for computing approximate gradients through back-propagation, as proposed by (Lee & Kim, 2023). White-box adaptive attacks, such as PGD+EOT, involve an iterative optimization process that requires computing the exact full gradients of the entire system, result in high memory usage and increased computational time. DBP methods often include a diffusion model as an add-on purifier, which the model requires extensive function calls during reverse generative process. Hence, it is hard to compute the exact full gradient of DBP systems efficiently. The surrogate process takes advantage of the fact that, given a fixed total amount of noise, we can denoise it using fewer denoising steps (Song et al., 2021a), but the gradients obtained from the surrogate process slightly differ from the exact gradients. Instead of using the full denoising steps, we approximate the original denoising process with fewer function calls, which allows us to compute gradients by directly back-propagating through the forward and reverse processes.

There are other gradient computation methods such as adjoint method in DiffPure (Li et al., 2020; Nie et al., 2022). It leverages an underlying numerical SDE solver to solve the reverse-time SDE. The adjoint method can theoretically compute exact full gradient, but in practice, it relies on the performance of the numerical solver, which is insufficient to measure the model robustness (Zhuang et al., 2020; Lee & Kim, 2023; Chen et al., 2024). Lee & Kim (2023) conducted a comprehensive evaluation of both gradient computation methods and concluded that utilizing the surrogate process for gradient computation poses a greater threat to model robustness. Hence, we use gradients obtained from a surrogate process in all our experiments.

## F ADAPTIVE WHITE-BOX PGD+EOT ATTACK FOR SSNI

---

**Algorithm 2** Adaptive white-box PGD+EOT attack for SSNI.

---

**Require:** clean data-label pairs $(\mathbf{x}, y)$; purifier $f_p$; classifier $f_c$; a noise level $T$; a score network $s_\theta(\mathbf{x}, T)$; objective function $\mathcal{L}$; perturbation budget $\epsilon$; step size $\alpha$; PGD iterations $K$; EOT iterations $N$.
1: Initialize $\mathbf{x}_0^{\text{adv}} \leftarrow \mathbf{x}$
2: **for** $k = 0, ..., K - 1$ **do**
3:     Computing sample-specific noise levels: $t(\mathbf{x}_k^{\text{adv}}) \leftarrow f(\left\| s_\theta(\mathbf{x}_k^{\text{adv}}, T) \right\|, T)$
4:     Average the gradients over EOT: $g_k \leftarrow \frac{1}{N} \sum_{i=1}^{N} \nabla_{\mathbf{x}^{\text{adv}}} \mathcal{L}\left(f_c(f_p(\mathbf{x}_k^{\text{adv}}, t(\mathbf{x}_k^{\text{adv}}))), y\right)$
5:     Update adversarial examples: $\mathbf{x}_{k+1}^{\text{adv}} \leftarrow \Pi_{\mathcal{B}_\epsilon(\mathbf{x})}\left(\mathbf{x}_k^{\text{adv}} + \alpha \cdot \text{sign}(g_k)\right)$
6: **end for**
7: return $\mathbf{x}^{\text{adv}} = \mathbf{x}_K^{\text{adv}}$

---

# G  ADAPTIVE WHITE-BOX BPDA+EOT ATTACK FOR SSNI

---

**Algorithm 3** Adaptive white-box BPDA+EOT attack.

---

**Require:** clean data-label pairs $(\mathbf{x}, y)$; purifier $f_p$; classifier $f_c$; approximation function $f_{\text{apx}}$; a noise level $T$; a score network $s_\theta(\mathbf{x}, T)$; objective function $\mathcal{L}$; perturbation budget $\epsilon$; step size $\alpha$; PGD iterations $K$; EOT iterations $N$.

1: Initialize $\mathbf{x}_0^{\text{adv}} \leftarrow \mathbf{x}$
2: **for** $k \leftarrow 0$ to $K - 1$ **do**
3:    Computing sample-specific $t$: $t(\mathbf{x}_k^{\text{adv}}) \leftarrow f(\left\| s_\theta(\mathbf{x}_k^{\text{adv}}, T) \right\|, T)$
4:    Average the gradient over EOT samples: $g_k \leftarrow \frac{1}{N} \sum_{i=1}^{N} \nabla_{\mathbf{x}^{\text{adv}}} \mathcal{L}\left( (f_c(f_{\text{apx}}(f_p(\mathbf{x}_k^{\text{adv}})))), y \right)$
5:    Update adversarial examples: $\mathbf{x}_{k+1}^{\text{adv}} \leftarrow \Pi_{\mathcal{B}_\epsilon(\mathbf{x})} \left( \mathbf{x}_k^{\text{adv}} + \alpha \cdot \text{sign}(g_k) \right)$
6: **end for**
7: **return** $\mathbf{x}^{\text{adv}} = \mathbf{x}_K^{\text{adv}}$

---

# H  PERFORMANCE EVALUATION OF SSNI-L

We also incorporate *SSNI-L* reweighting framework with existing DBP methods for accuracy-robustness evaluation. In Table 5, we report the results against PGD+EOT $\ell_\infty(\epsilon = 8/255)$ and $\ell_2(\epsilon = 0.5)$ threat models on CIFAR-10, respectively. We can see that *SSNI-L* can still support DBP methods to better trade-off between standard accuracy and robust accuracy. Also, we report the results against BPDA+EOT $\ell_\infty(\epsilon = 8/255)$ threat model on CIFAR-10 in Table 6. Overall, *SSNI-L* slightly decreases the robustness of DBP methods against PGD+EOT attack and maintain the robustness of DBP methods against BPDA+EOT attack, but there is a notable improvement in standard accuracy.

Table 5: Standard and robust accuracy of DBP methods against adaptive white-box PGD+EOT (left: $\ell_\infty(\epsilon = 8/255)$, right: $\ell_2(\epsilon = 0.5)$) on *CIFAR-10*. WideResNet-28-10 and WideResNet-70-16 are used as classifiers. We compare the result of DBP methods with and without *SSNI-L*. We report mean and standard deviation over three runs. We show the most successful defense in **bold**.

| | PGD+EOT $\ell_\infty$ ($\epsilon = 8/255$) | | | | PGD+EOT $\ell_2$ ($\epsilon = 0.5$) | | |
|---|---|---|---|---|---|---|---|
| | DBP Method | Standard | Robust | | DBP Method | Standard | Robust |
| WRN-28-10 | Nie et al. (2022) | 89.71±0.72 | **47.98±0.64** | WRN-28-10 | Nie et al. (2022) | 91.80±0.84 | **82.81±0.97** |
| | + *SSNI-L* | **92.97±0.42** | 46.35±0.72 | | + *SSNI-L* | **93.82±0.37** | 81.12±0.80 |
| | Wang et al. (2022) | 92.45±0.64 | **36.72±1.05** | | Wang et al. (2022) | 92.45±0.64 | **82.29±0.82** |
| | + *SSNI-L* | **93.62±0.49** | 36.59±1.29 | | + *SSNI-L* | **93.62±0.49** | 80.66±1.31 |
| | Lee & Kim (2023) | 90.1±0.18 | **56.05±1.11** | | Lee & Kim (2023) | 90.10±0.18 | 83.66±0.46 |
| | + *SSNI-L* | **93.49±0.33** | 53.71±0.48 | | + *SSNI-L* | **93.49±0.33** | **85.29±0.24** |
| WRN-70-16 | Nie et al. (2022) | 90.89±1.13 | **52.15±2.30** | WRN-70-16 | Nie et al. (2022) | 92.90±0.40 | 82.94±1.13 |
| | + *SSNI-L* | **93.82±0.49** | 49.94±0.33 | | + *SSNI-L* | **94.99±0.24** | **84.44±0.56** |
| | Wang et al. (2022) | 93.10±0.51 | **43.55±0.58** | | Wang et al. (2022) | 93.10±0.51 | **85.03±0.49** |
| | + *SSNI-L* | **93.88±0.49** | 43.03±0.60 | | + *SSNI-L* | **93.88±0.49** | 82.88±0.79 |
| | Lee & Kim (2023) | 89.39±1.12 | **56.97±0.33** | | Lee & Kim (2023) | 89.39±1.12 | 84.51±0.37 |
| | + *SSNI-L* | **92.64±0.40** | 52.86±0.46 | | + *SSNI-L* | **92.64±0.40** | **84.90±0.09** |

Table 6: Standard and robust accuracy (%) against adaptive white-box BPDA+EOT $\ell_\infty (\epsilon = 8/255)$ attack on *CIFAR-10*. We compare the result of DBP methods with and without *SSNI-L*. We report mean and standard deviation over three runs. We show the most successful defense in **bold**.

| | BPDA+EOT $\ell_\infty$ ($\epsilon = 8/255$) | | |
|---|---|---|---|
| | DBP Method | Standard | Robust |
| WRN-28-10 | Nie et al. (2022) | 89.71±0.72 | **81.90±0.49** |
| | + *SSNI-L* | **92.97±0.42** | 80.08±0.96 |
| | Wang et al. (2022) | 92.45±0.64 | 79.88±0.89 |
| | + *SSNI-L* | **93.62±0.49** | **79.95±1.12** |
| | Lee & Kim (2023) | 90.10±0.18 | 88.40±0.88 |
| | + *SSNI-L* | **93.49±0.33** | **88.41±0.09** |

# I  ABLATION STUDY ON SCORE NORM

We further provide experiments with single score norm as reweighting metric instead of EPS norm. Yoon et al. (2021) shows score norm $\nabla_{\mathbf{x}} \log p_t(\mathbf{x})$ is a valid measurement for adversarial detection. Incorporating single score norm with our SSNI-N framework, it still achieves notable improvement on standard accuracy, but the robustness of the DBP methods is exhausted. The single score norm is highly sensitive to the noise levels, which makes it insufficient to completely distinguish between natural and adversarial examples.

Table 7: Standard and robust accuracy (%) against adaptive white-box PGD+EOT $\ell_\infty(\epsilon = 8/255)$ attack on *CIFAR-10*. We use *single score norms* (i.e., $\|\nabla_{\mathbf{x}} \log p_t(\mathbf{x})\|$). We report mean and standard deviation over three runs. We show the most successful defense in **bold**.

| | PGD+EOT $\ell_\infty$ ($\epsilon = 8/255$) | | |
|---|---|---|---|
| | DBP Method | Standard | Robust |
| WRN-28-10 | Nie et al. (2022) | 89.71±0.72 | **47.98±0.64** |
| | + *SSNI-N* | **92.84±0.18** | 47.20±1.22 |
| | Wang et al. (2022) | 92.45±0.64 | **36.72±1.05** |
| | + *SSNI-N* | **93.42±0.60** | 34.24±1.45 |
| | Lee & Kim (2023) | 90.10±0.18 | **56.05±1.11** |
| | + *SSNI-N* | **93.55±0.42** | 55.47±1.15 |

# J  ABLATION STUDY ON BIAS TERM

Table 8: Ablation study on the bias term $b$. We report the standard and robust accuracy of DBP methods against adaptive white-box PGD+EOT on *CIFAR-10*. WideResNet-28-10 and WideResNet-70-16 are used as classifiers. We report mean and standard deviation over three runs. We show the most successful defense in **bold**.

| | PGD+EOT $\ell_\infty$ ($\epsilon = 8/255$) | | | | | | |
|---|---|---|---|---|---|---|---|
| Bias | 0 | 5 | 10 | 15 | 20 | 25 | 30 |
| | WRN-28-10 | | | | | | |
| Standard | **94.34 ± 1.43** | 93.95 ± 1.17 | 93.17 ± 1.05 | 92.38 ± 1.29 | 92.38 ± 1.29 | 92.10 ± 1.03 | 92.97 ± 0.37 |
| Robust | 55.27 ± 0.75 | 57.23 ± 1.62 | 57.03 ± 0.54 | 57.64 ± 0.89 | 57.64 ± 0.89 | 58.24 ± 1.22 | **59.18 ± 1.65** |
| | WRN-70-16 | | | | | | |
| Standard | 94.34 ± 0.45 | 93.82 ± 1.56 | **94.73 ± 1.34** | 92.79 ± 0.89 | 92.79 ± 0.89 | 92.58 ± 0.67 | 92.77 ± 0.94 |
| Robust | 56.45 ± 1.22 | 57.03 ± 0.78 | 58.10 ± 0.24 | 58.79 ± 1.48 | **59.57 ± 1.19** | 58.59 ± 0.52 | 58.59 ± 0.52 |

Table 9: Ablation study on the bias term $b$. We report standard and robust accuracy of DBP methods against adaptive white-box PGD+EOT on *ImageNet-1K*. ResNet-50 is used as the classifier. We report mean and standard deviation over three runs. We show the most successful defense in **bold**.

| PGD+EOT $\ell_\infty$ ($\epsilon = 4/255$) | | | | | | | |
|---|---|---|---|---|---|---|---|
| Bias | 0 | 25 | 50 | 75 | 100 | 125 | 150 |
| RN-50 | | | | | | | |
| Standard | $71.68 \pm 1.12$ | $71.73 \pm 1.49$ | $\mathbf{71.96 \pm 0.13}$ | $68.80 \pm 0.74$ | $68.41 \pm 0.59$ | $67.63 \pm 1.08$ | $66.45 \pm 1.53$ |
| Robust | $39.33 \pm 0.34$ | $40.28 \pm 0.28$ | $\mathbf{43.88 \pm 0.22}$ | $41.45 \pm 0.38$ | $43.02 \pm 0.41$ | $40.87 \pm 0.92$ | $40.05 \pm 0.67$ |

## K  COMPUTE RESOURCES

Table 10: Inference time of the DBP methods with and without SSNI for a single image running on one A100 GPU on *CIFAR-10*. We use WideResNet-28-10 as the classifier.

| DBP Method | Reweighting Method | Time (s) |
|---|---|---|
| Nie et al. (2022) | - | 3.934 |
| | SSNI-L | 4.473 |
| | SSNI-N | 4.474 |
| Wang et al. (2022) | - | 5.174 |
| | SSNI-L | 5.793 |
| | SSNI-N | 5.829 |
| Lee & Kim (2023) | - | 14.902 |
| | SSNI-L | 15.624 |
| | SSNI-N | 15.534 |

Table 11: Inference time of the DBP methods with and without SSNI for a single image running on one A100 GPU on *ImageNet-1K*. We use ResNet-50 as the classifier.

| DBP Method | Reweighting Method | Time (s) |
|---|---|---|
| Nie et al. (2022) | - | 8.980 |
| | SSNI-L | 14.515 |
| | SSNI-N | 14.437 |
| Wang et al. (2022) | - | 11.271 |
| | SSNI-L | 16.657 |
| | SSNI-N | 16.747 |
| Lee & Kim (2023) | - | 35.091 |
| | SSNI-L | 40.526 |
| | SSNI-N | 40.633 |

We implemented our code on Python version 3.8, CUDA version 12.2.0, and PyTorch version 2.0.1 with Slurm Workload Manager. We conduct each of the experiments on up to $4 \times$ NVIDIA A100 GPUs. Source code for this work is available at: https://anonymous.4open.science/r/SSNI-F746.

# L  VISUALIZATIONS FOR PURIFICATION RESULTS

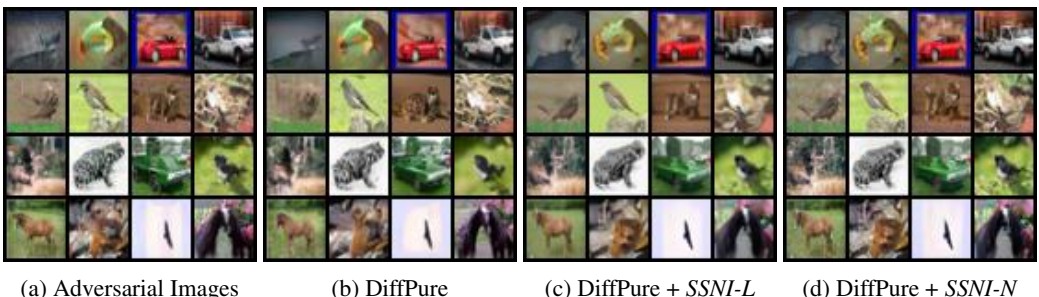

(a) Adversarial Images      (b) DiffPure      (c) DiffPure + *SSNI-L*      (d) DiffPure + *SSNI-N*

Figure 5: More results on purification effects.

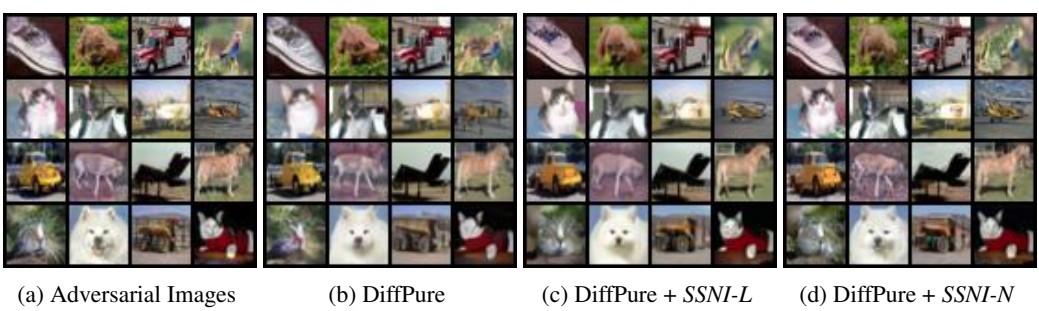

(a) Adversarial Images      (b) DiffPure      (c) DiffPure + *SSNI-L*      (d) DiffPure + *SSNI-N*

Figure 6: More results on purification effects.

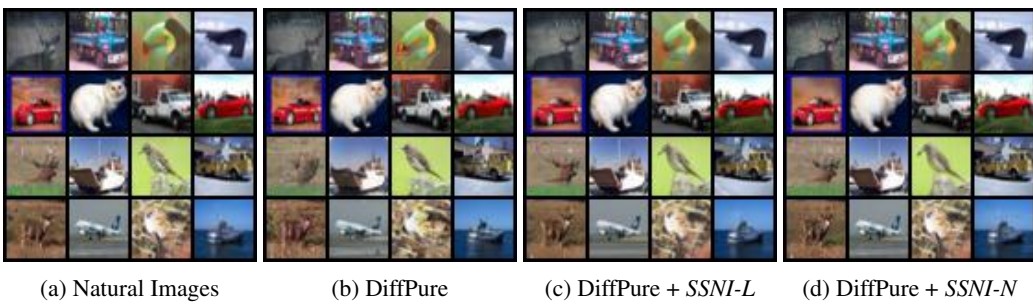

(a) Natural Images      (b) DiffPure      (c) DiffPure + *SSNI-L*      (d) DiffPure + *SSNI-N*

Figure 7: More results on purification effects.

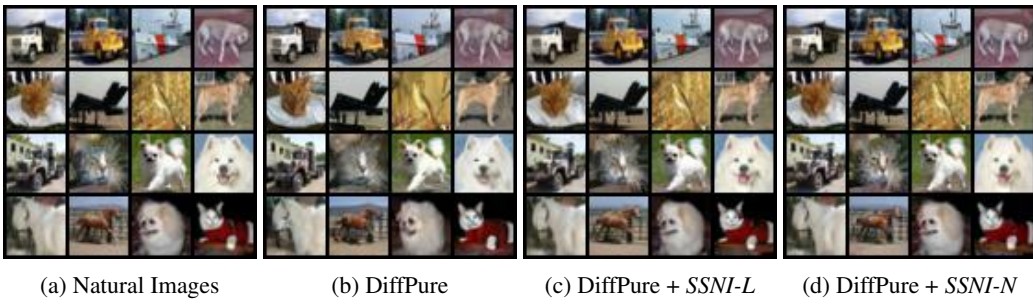

(a) Natural Images      (b) DiffPure      (c) DiffPure + *SSNI-L*      (d) DiffPure + *SSNI-N*

Figure 8: More results on purification effects.

