# OpenReview forum: "Sample-specific Noise Injection for Diffusion-based Adversarial Purification"
_ICLR.cc/2025/Conference — Submitted to ICLR 2025_

### Official Review · Reviewer_iJtf · 2024-10-24

**Soundness:** 2
**Presentation:** 2
**Contribution:** 2
**Rating:** 6
**Confidence:** 4

**Summary:**

In this paper, the authors proposed to purify the adversarial examples via diffusion models with sample specific noise levels. Hence the adversarial purification can both sufficiently remove the adversarial attacks and preserve the semantic information. In order to obtain the sample-specific noise levels, the authors started from their other key finding that the difference between the clean image and the data point in the purification process can be measured using the score norm. In their experiments, the proposed method SSNI demonstrated both higher standard accuracy and robust accuracy than the baseline diffusion-based purification methods

**Strengths:**

1. *Originality*: It is an original approach to measure the difference between the clean data and the data point sampled from the reverse process in the diffusion model, using the score norm. Given this score norm, it is also novel to compute sample specific noise levels for purification on the adversarial examples.

2. *Clarity*: The core method sample-specific score-aware noise injection is clearly articulated. It is easy to follow from the framework of the method to the design of experiments. The key contributions are also explicitly presented with high clarity.

**Weaknesses:**

1. *Presentation*: The presentation of this work should be improved, especially for the visual presentation. For example, Figure 1 would be clearer to deliver the observations using the ImageNet dataset, because it has higher-resolution images which allow the audience to distinguish the objects.


2. *Quality*: I would highly recommend the authors to have a sufficient check on their lemmas and propositions, especially the proofs. Based on the coherent and flawless proofs, the lemmas and the propositions would be more convincing and solid. For example, the plus or minus sign in the proof of Proposition 1 when they apply the Cauchy-Schwarz inequality. More examples can be seen in **Questions**.

**Questions:**

1. In terms of the significance and the novelty of this paper, could the authors compare the novelty of this work with Diffusion Models Demand Contrastive Guidance for Adversarial Purification to Advance [1] in ICML 2024 in terms of the $t^*$ selection? It would be better to present the difference to highlight the contributions in this submitted paper.

2. For Lemma 3, there is a minor question on in the proof: it seems that given $K>0$, $(1 - \epsilon - K\epsilon) || \mathbf{x} || \geq (1-2\epsilon) ||\mathbf{x}||$ may not hold. Would the authors please elaborate it?

3. For Proposition 1, could the authors explain why the derivative of $\bar{\alpha}_t$ is $\frac{\partial \bar{\alpha}_t}{\partial t} = -\beta_t \bar{\alpha}_t$, given $\bar{\alpha}_t= \prod^t (1-\beta_t)$?

4. BPDA+EOT as a white-box attack has been shown to achieve lower attack success rate than those utilizing direct gradients of the defense process when evaluating diffusion-based purification methods [2]. It would be better to elaborate the reason why BPDA+EOT is chosen to evaluate the proposed diffusion-based purification method.


[1] Bai, M., Huang, W., Li, T., Wang, A., Gao, J., Caiafa, C.F. & Zhao, Q.. (2024). Diffusion Models Demand Contrastive Guidance for Adversarial Purification to Advance, Proceedings of the 41st International Conference on Machine Learning.

[2] Minjong Lee and Dongwoo Kim. Robust evaluation of diffusion-based adversarial purification. In ICCV, 2023.

---

> ### Author Response · Authors · 2024-11-22
> **Response to Reviewer iJtf (part 1)**
>
> Thank you so much for your comments! It is our pleasure that the novelty of our method and the overall clarity can be recognized. Your thorough review and comments are very important to the improvement of our work! **Please find our replies below.**
>
> >W1. Presentation: The presentation of this work should be improved, especially for the visual presentation. For example, Figure 1 would be clearer to deliver the observations using the ImageNet dataset, because it has higher-resolution images which allow the audience to distinguish the objects.
>
> **Reply:** Thanks for your suggestion! We **agree** with you that ImageNet-1K images are of higher resolution, which could make objects more distinguishable. However, the high resolution of ImageNet images, combined with the relatively small magnitude of adversarial noise, results in **less visible differences** between purified samples at different $t^*$ values compared to CIFAR-10. In our manuscript, Figure 1b can clearly demonstrate how the dog's semantics are purified into a frog's semantics, finally leading to misclassification. The aim of Figure 1 is to showcase such visible semantic shifts and highlight the motivation behind our method. However, unfortunately, using ImageNet-1K will **make the semantic shifts less visible**.
>
> >W2. Quality: I would highly recommend the authors to have a sufficient check on their lemmas and propositions, especially the proofs. Based on the coherent and flawless proofs, the lemmas and the propositions would be more convincing and solid. For example, the plus or minus sign in the proof of Proposition 1 when they apply the Cauchy-Schwarz inequality. More examples can be seen in Questions.
>
> **Reply:** Thank you for your thorough review and for highlighting these concerns.
> We have carefully revised the proofs of the Lemma and Proposition in our paper. All of the updated content is highlighted in blue.
> Below we summarize the key fixes to the specific examples as you mentioned:
>
> **Cauchy-Schwarz inequality**
>
> The change from the minus to the plus sign, i.e., from
> $$
> | \frac{\partial g}{\partial t} | \leq \frac{ 2\beta\_t \bar{\alpha}\_t}{(1-\bar{\alpha}\_t)^2} \left\| s\_t (\mathbf{x}) \right\| \cdot \left\| \mathbf{x} - \frac{1+\bar{\alpha}\_t}{2\sqrt{\bar{\alpha}\_t}}\mathbb{E}_{p(\mathbf{x}\_0|\mathbf{x})}[\mathbf{x}\_0] \right\|
> $$
> to
>
> $$
>    | \frac{\partial g}{\partial t} | \leq \frac{2\beta\_t\bar{\alpha}\_t}{(1-\bar{\alpha}\_t)^2} \left\| s\_t(\mathbf{x}) \right\| \cdot \left( \left\| \mathbf{x} \right\| + \left\| \frac{1+\bar{\alpha}\_t}{2\sqrt{\bar{\alpha}\_t}}\mathbb{E}\_{p(\mathbf{x}_0|\mathbf{x})}[\mathbf{x}\_0] \right\| \right)
> $$
> comes as the result of applying triangle inequality:
>
> $$
> || \mathbf{a} - \mathbf{b} || = || \mathbf{a} + (-\mathbf{b}) || \leq || \mathbf{a} || + || -\mathbf{b} || = || \mathbf{a} || + || \mathbf{b} ||
> $$
>
> *The derivation is fully detailed in the revised manuscript accordingly.*
>
> >Q1. In terms of the significance and the novelty of this paper, could the authors compare the novelty of this work with Diffusion Models Demand Contrastive Guidance for Adversarial Purification to Advance [1] in ICML 2024 in terms of the t^* selection? It would be better to present the difference to highlight the contributions in this submitted paper. [1] Bai, M., Huang, W., Li, T., Wang, A., Gao, J., Caiafa, C.F. & Zhao, Q.. (2024). Diffusion Models Demand Contrastive Guidance for Adversarial Purification to Advance, Proceedings of the 41st International Conference on Machine Learning.
>
> **Reply:** Thank you for bringing this interesting paper to our attention! We summarize some key differences between [1] and our work here:
> - [1] aims to improve the performance of diffusion-based purification methods by guiding diffusion models with contrastive loss **through the reverse process**. In terms of the $t^*$ selection, [1] uses a constant $t^*$ for all samples.
> - On the other hand, our method focuses on sample-specific reweighting on timestep $t^*$ (i.e., different sample may have different $t^*$), leading to different amounts of Gaussian noise being injected into the samples **in the forward process**.
>
> **For more detailed discussions, we will add a paragraph in the updated version of our manuscript!**
>
>  [1] Diffusion Models Demand Contrastive Guidance for Adversarial Purification to Advance, ICML 2024.
>
> >Q2. For Lemma 3, there is a minor question on in the proof: it seems that given $K > 0, (1 - \epsilon - K\epsilon)||\mathbf{x}|| \geq (1 - 2\epsilon)||\mathbf{x}||$ may not hold. Would the authors please elaborate on it?
>
> **Reply:** We are very grateful for pointing this out. This was indeed a flaw. The condition that $\epsilon$ should satisfy has been updated to finding the discriminant of the desired inequality $||s_t(\mathbf{x})|| \geq C || \mathbf{x} ||$. We **have included** the fix in the revision. Please kindly refer to Appendix A of the newly uploaded PDF.

---

> ### Author Response · Authors · 2024-11-22
> **Response to Reviewer iJtf (part 2)**
>
> >Q3. For Proposition 1, could the authors explain why the derivative of $\bar{\alpha}$ is $\frac{\partial \bar{\alpha}_t}{\partial t} = -\beta_t \bar{\alpha}_t$, given $\bar{\alpha}_t = \prod^t(1-\beta_t)?$
>
> **Reply:** Thank you for your question. We expanded the details of the approximations and simplifications used in deriving this derivative as follows:
>
> Given that $\bar{\alpha}\_t = \prod\_{i=1}^{t} (1 - \beta\_i)$, we first consider a continuous approximation of $t$, where the product is approximated as an exponential of the integral
> $$
> \bar{\alpha}\_t = \exp \left( \int\_{0}^{t} \log ( 1 - \beta_i) \mathrm{d} i \right)
> $$
> and apply first-order Taylor approximation to $\beta\_t$ (valid under small $\beta\_t$ values in common diffusion model practices), i.e., $\log( 1 - \beta\_i) \approx - \beta\_i$, leading to the approximated $\bar{a}\_t \approx \exp \left( - \int\_0^t \beta\_i \mathrm{d} i \right)$. This way we compute the derivative of $\bar{a}_t$ as
> $$
>         \begin{aligned}
>             \frac{\partial }{\partial t} \bar{a}_t & = \frac{\partial }{\partial t} \exp \left( - \int_0^t \beta_i \mathrm{d} i \right) \\\\
>                 & = \exp \left( - \int_0^t \beta_i \mathrm{d} i \right)  \frac{\partial }{\partial t} \left( - \int_0^t \beta_i \mathrm{d} i \right) \\\\
>                 & = \exp \left( - \int_0^t \beta_i \mathrm{d} i \right) (-\beta_t) \\\\
>                 & = - \beta_t \bar{\alpha}_t
>         \end{aligned}
> $$
>
> This expanded derivation, along with subsequent details, has been incorporated into the revised manuscript for completeness. To keep the response concise and targeted, we do not include them in this message.
>
> We deeply appreciate your careful reading and insightful comments, which have allowed us to improve the rigor and clarity of our submission. We hope this response and the revised manuscript address your concerns.
>
> >Q4. BPDA+EOT as a white-box attack has been shown to achieve lower attack success rate than those utilizing direct gradients of the defense process when evaluating diffusion-based purification methods [2]. It would be better to elaborate the reason why BPDA+EOT is chosen to evaluate the proposed diffusion-based purification method. [2] Minjong Lee and Dongwoo Kim. Robust evaluation of diffusion-based adversarial purification. In ICCV, 2023.
>
> **Reply:** Thanks for your insightful comments!  We would like to point out that our experiments follow the settings in [1], and our **main results are presented under the evaluations of adaptive white-box PGD+EOT.** The main reason for using BPDA+EOT is to make our experiments **more comprehensive and to ensure diversity in the attacks** (i.e., evaluate the robustness of our proposed framework under different attack scenarios). Specifically, BPDA+EOT and PGD+EOT attacks differ fundamentally in their methodologies. BPDA uses an identity function to estimate gradients, which can significantly increase the evaluation time for DBP methods. In contrast, PGD+EOT computes full backpropagation gradients, which can result in memory exhaustion due to high computational demands.
>
> [1] Robust Evaluation of Diffusion-Based Adversarial Purification, ICCV 2023.

---

> ### Author Response · Authors · 2024-11-22
> **Response to Reviewer iJtf**
>
> **In the end,** we want to thank you for providing this valuable feedback to us, which is always important to hear opinions from other experts. If you feel there are still unclear points regarding our paper, **please discuss** with us in the author-reviewer discussion phase. If you feel that your concerns are well-addressed, we do hope for an updated rating and new comments (if possible). Thanks again!

---

> > ### Comment · Reviewer_iJtf · 2024-11-25
> >
> > I thank the authors for their careful response to my questions and comments. All my questions are addressed. I would increase my score by 1.

---

> > > ### Author Response · Authors · 2024-11-25
> > > **Many thanks for your reply and increasing your score to 6!**
> > >
> > > Dear Reviewer iJtf,
> > >
> > > Many thanks for your valuable comments again, and we are glad to hear that your concerns are well addressed.
> > >
> > > Best regards,
> > >
> > > Authors of Submission 10115

---

> > > > ### Author Response · Authors · 2024-11-26
> > > >
> > > > Dear Reviewer iJtf,
> > > >
> > > > Many thanks for your valuable comments again!
> > > >
> > > > Since the current recommendation is borderline instead of "a good paper", we are not sure if there is anything we can do to make our paper even better? We can further strengthen our paper based on your new comments until our paper is good to be accepted. ^^
> > > >
> > > > We also want to share with you the current situation regarding the other reviews. Unfortunately, Reviewer mLed gave a rejection recommendation despite acknowledging that we addressed all the concerns, and another reviewer (Reviewer LRdc) has not yet to provide any response at all.
> > > >
> > > > This has placed our submission in a particularly challenging position (i.e., the averaged rating is around borderline). We would be deeply grateful for your consideration of further raising your confidence or score. Your support would play a crucial role in ensuring our paper can be fairly evaluated in this rebuttal.
> > > >
> > > > Looking forward to hearing back from you.
> > > >
> > > > Best regards,
> > > >
> > > > Authors of Submission 10115

---

> > > ### Author Response · Authors · 2024-12-03
> > > **Official Comment by Authors**
> > >
> > > Dear Reviewer iJtf,
> > >
> > > Many thanks for your valuable comments again!
> > >
> > > Since the current recommendation is borderline instead of "a good paper", we are not sure if there is anything we can do to make our paper even better? We can further strengthen our paper based on your new comments until our paper is good to be accepted. ^^
> > >
> > > We also want to share with you the current situation regarding the other reviews. Unfortunately, Reviewer mLed gave a rejection recommendation despite acknowledging that we addressed all the concerns.
> > >
> > > This has placed our submission in a particularly challenging position (i.e., the averaged rating is around borderline). We would be deeply grateful for your consideration of further raising your confidence or score. Your support would play a crucial role in ensuring our paper can be fairly evaluated in this rebuttal.
> > >
> > > Looking forward to hearing back from you.
> > >
> > > Best regards,
> > >
> > > Authors of Submission 10115

---

> ### Author Response · Authors · 2024-11-23
> **Reminder - Discussion Stage Closing Soon - 23 November**
>
> Dear Reviewer iJtf,
>
> We appreciate the time and effort that you have dedicated to reviewing our manuscript.
>
> We have carefully addressed all your queries. Could you kindly spare a moment to review our responses?
>
> Have our responses addressed your major concerns?
>
> If there is anything unclear, we will address it further. We look forward to your feedback.
>
> Best regards,
>
> Authors of Submission 10115

---

> ### Author Response · Authors · 2024-11-24
> **Reminder - Discussion Stage Closing Soon - 24 November**
>
> Dear Reviewer iJtf,
>
> We appreciate the time and effort that you have dedicated to reviewing our manuscript.
>
> We have carefully addressed all your queries. Could you kindly spare a moment to review our responses?
>
> Have our responses addressed your major concerns?
>
> If there is anything unclear, we will address it further. We look forward to your feedback.
>
> Best regards,
>
> Authors of Submission 10115

---

> ### Author Response · Authors · 2024-11-25
> **Reminder - Discussion Stage Closing Soon - 25 November**
>
> Dear Reviewer iJtf,
>
> We appreciate the time and effort that you have dedicated to reviewing our manuscript.
>
> We have carefully addressed all your queries. Could you kindly spare a moment to review our responses?
>
> Have our responses addressed your major concerns?
>
> If there is anything unclear, we will address it further. We look forward to your feedback.
>
> Best regards,
>
> Authors of Submission 10115

---

> ### Author Response · Authors · 2024-12-02
>
> Dear Reviewer iJtf,
>
> Many thanks for your valuable comments again!
>
> Since the current recommendation is borderline instead of "a good paper", we are not sure if there is anything we can do to make our paper even better? We can further strengthen our paper based on your new comments until our paper is good to be accepted. ^^
>
> We also want to share with you the current situation regarding the other reviews. Unfortunately, Reviewer mLed gave a rejection recommendation despite acknowledging that we addressed all the concerns.
>
> This has placed our submission in a particularly challenging position (i.e., the averaged rating is around borderline). We would be deeply grateful for your consideration of further raising your confidence or score. Your support would play a crucial role in ensuring our paper can be fairly evaluated in this rebuttal.
>
> Looking forward to hearing back from you.
>
> Best regards,
>
> Authors of Submission 10115

---

### Official Review · Reviewer_mLed · 2024-11-03

**Soundness:** 3
**Presentation:** 3
**Contribution:** 3
**Rating:** 5
**Confidence:** 3

**Summary:**

The work provide a method to adapt the diffusion step for each sample in adversarial purification based on diffusion model in order to enhance the trade-off between robust acc and test acc. The paper use the norm of score to reflect how much a data point deviates from the clean data distribution and utilize the metric to design the diffusion step function. The work show a theoretical explanation that the score norm can act as a proxy for distinguishing between noise levels.

**Strengths:**

Good theory and experimental results

**Weaknesses:**

None

**Questions:**

How to adapt bias in the reweighting function in different datasets and network ?

---

> ### Author Response · Authors · 2024-11-22
> **Response to Reviewer mLed**
>
> Thank you so much for your comments! It is our pleasure that the theoretical justification and the experiment results can be recognized. Your thorough review and comments are very important to the improvement of our work! **Please find our replies below.**
>
> >Q1. How to adapt bias in the reweighting function in different datasets and network?
>
> **Reply:**
> - **Justification:** The design principle of the reweighting function is to scale the timestep $t^*$ based on the score information. As we have observed that clean data often has a much lower norm value than adversarial one, we believe that incorporating a linear function or a sigmoid function would reweight the timestep properly. For realizations, we extract 5,000 validation clean examples from the training data (denoted as  $\mathbf{x}_v$) and we use $\left\|\text{EPS}(\mathbf{x}_v)\right\|$ as a reference to indicate the approximate EPS norm values of clean data, which can help us reweight $t^*$. Thus, based on the EPS norm information, the reweighting function assigns adaptive timestep accordingly.
> - **How to select bias term b:** When we select the bias term, we first split a validation set from each dataset and record the performance in above tables. We then choose the value based on the validation performance, then we use the term in our test data evaluations.
>
> We apologize that our manuscript lacks of ablation study of the bias term. We present the result below. The bias term has the same magnitude with the timestep $t^*$. Also, there is a small mistake in Equation 9, the correct formula should be:
> $f_\sigma(\left\|\text{EPS}(\mathbf{x})\right\|,~t^*) = \frac{t^* + b}{1 + \exp\{-(\left\|\text{EPS}(\mathbf{x})\right\|-\mu)/\tau\}}$
>
> Table 1: Ablation study on the bias term $b$. We report clean and robust accuracy (%) against white-box PGD+EOT adaptive attacks ($\ell_\infty, \epsilon=8/255$) on CIFAR-10. The classifier used is WRN-28-10.
> |Classifier|Bias|Standard|Robust|
> |--|--|--|--|
> |WRN-28-10|0|**94.34$\pm$ 1.43**|55.27$\pm$ 0.75|
> |WRN-28-10|5|93.95$\pm$ 1.17|57.23$\pm$ 1.62|
> |WRN-28-10|10|93.17$\pm$ 1.05|57.03$\pm$ 0.54|
> |WRN-28-10|15|92.38$\pm$ 1.29|57.64$\pm$ 0.89|
> |WRN-28-10|20|92.38$\pm$ 1.29|57.64$\pm$ 0.89|
> |WRN-28-10|25|92.10$\pm$ 1.03|58.24$\pm$ 1.22|
> |WRN-28-10|30|92.97$\pm$ 0.37|**59.18$\pm$ 1.65**|
>
>
> Table 2: Ablation study on the bias term $b$. We report clean and robust accuracy (%) against white-box PGD+EOT adaptive attacks ($\ell_\infty, \epsilon=8/255$) on CIFAR-10. The classifier used is WRN-70-16.
> |Classifier|Bias|Standard|Robust|
> |--|--|--|--|
> |WRN-70-16|0|94.34$\pm$ 0.45|56.45$\pm$ 1.22|
> |WRN-70-16|5|93.82$\pm$ 1.56|57.03$\pm$ 0.78|
> |WRN-70-16|10|**94.73$\pm$ 1.34**|58.10$\pm$ 0.24|
> |WRN-70-16|15|92.97$\pm$ 0.89|58.79$\pm$ 1.48|
> |WRN-70-16|20|92.97$\pm$ 0.89|**59.57$\pm$ 1.19**|
> |WRN-70-16|25|92.58$\pm$ 0.67|58.59$\pm$ 0.52|
> |WRN-70-16|30|92.77$\pm$ 0.94|58.59$\pm$ 0.52|
>
>
> Table 3: Ablation study on the reweighting function parameter $b$. We report clean and robust accuracy (%) against white-box PGD+EOT adaptive attacks ($\ell_\infty, \epsilon=4/255$) on ImageNet-1K. The classifier used is ResNet-50.
> |Classifier|Bias|Standard|Robust|
> |--|--|--|--|
> |ResNet-50|0|71.68$\pm$ 1.12|39.93$\pm$ 0.34|
> |ResNet-50|25|71.73$\pm$ 1.49|40.28$\pm$ 0.28|
> |ResNet-50|50|**71.96$\pm$ 0.13**|**43.88$\pm$ 0.22**|
> |ResNet-50|75|68.80$\pm$ 0.74|41.45$\pm$ 0.38|
> |ResNet-50|100|68.41$\pm$ 0.59|43.02$\pm$ 0.41|
> |ResNet-50|125|67.63$\pm$ 1.08|40.87$\pm$ 0.92|
> |ResNet-50|150|66.45$\pm$ 1.53|40.05$\pm$ 0.67|
>
> From the experimental results, we can derive the following conclusions: the selection of bias term does not impact the performance of our framework under CIFAR-10 and ImageNet-1K. Note that when bias increases, there is a general observation that the clean accuracy drops and the robust accuracy increases. This perfectly aligns with the understanding of optimal timestep $t^*$ selection in the previous DBP work, where large timestep would lead to drop of both clean and robust accuracy and small timestep cannot remove the adversarial perturbation effectively (see Figure 1 in [1]).
>
> [1] Robust Evaluation of Diffusion-Based Adversarial Purification, ICCV 2023.

---

> ### Author Response · Authors · 2024-11-22
> **Response to Reviewer mLed**
>
> **In the end,** we want to thank you for providing this valuable feedback to us, which is always important to hear opinions from other experts. If you feel there are still unclear points regarding our paper, **please discuss** with us in the author-reviewer discussion phase. If you feel that your concerns are well-addressed, we do hope for an updated rating and new comments (if possible). Thanks again!

---

> ### Author Response · Authors · 2024-11-23
> **Reminder - Discussion Stage Closing Soon - 23 November**
>
> Dear Reviewer mLed,
>
> We appreciate the time and effort that you have dedicated to reviewing our manuscript.
>
> We have carefully addressed all your queries. Could you kindly spare a moment to review our responses?
>
> Have our responses addressed your major concerns?
>
> If there is anything unclear, we will address it further. We look forward to your feedback.
>
> Best regards,
>
> Authors of Submission 10115

---

> ### Author Response · Authors · 2024-11-24
> **Reminder - Discussion Stage Closing Soon - 24 November**
>
> Dear Reviewer mLed,
>
> We appreciate the time and effort that you have dedicated to reviewing our manuscript.
>
> We have carefully addressed all your queries. Could you kindly spare a moment to review our responses?
>
> Have our responses addressed your major concerns?
>
> If there is anything unclear, we will address it further. We look forward to your feedback.
>
> Best regards,
>
> Authors of Submission 10115

---

> ### Author Response · Authors · 2024-11-25
> **Reminder - Discussion Stage Closing Soon - 25 November**
>
> Dear Reviewer mLed,
>
> We appreciate the time and effort that you have dedicated to reviewing our manuscript.
>
> We have carefully addressed all your queries. Could you kindly spare a moment to review our responses?
>
> Have our responses addressed your major concerns?
>
> If there is anything unclear, we will address it further. We look forward to your feedback.
>
> Best regards,
>
> Authors of Submission 10115

---

> > ### Comment · Reviewer_mLed · 2024-11-26
> >
> > Thank you for your response. All my questions are addressed. I keep the rating unchanged.

---

> > > ### Author Response · Authors · 2024-11-26
> > >
> > > Dear Reviewer mLed,
> > >
> > > Many thanks for your reply. It is glad to hear that your concerns have been addressed.
> > >
> > > If you feel that your concerns have been well-addressed, we do hope for an updated rating, as it would be a strong encouragement for our efforts.
> > >
> > > We noticed that the current score remains "borderline reject," despite **addressing all your concerns and no weaknesses being mentioned**. Could you kindly clarify **what prevents you from assigning a more positive score?** Your additional insights would be invaluable for us to further refine our work. There is still time to discuss, we are delighted to answer any questions or address any concerns from you.
> > >
> > > We are looking forward to your reply!
> > >
> > > Best regards,
> > >
> > > Authors of Submission 10115

---

> > > > ### Author Response · Authors · 2024-12-02
> > > >
> > > > Dear Reviewer mLed,
> > > >
> > > > Many thanks for your reply. It is glad to hear that your concerns have been addressed.
> > > >
> > > > If you feel that your concerns have been well-addressed, we do hope for an updated rating, as it would be a strong encouragement for our efforts.
> > > >
> > > > We noticed that the current score remains "borderline reject," despite **addressing all your concerns and no weaknesses being mentioned**. Could you kindly clarify **what prevents you from assigning a more positive score?** Your additional insights would be invaluable for us to further refine our work. There is still time to discuss, we are delighted to answer any questions or address any concerns from you.
> > > >
> > > > We are looking forward to your reply!
> > > >
> > > > Best regards,
> > > >
> > > > Authors of Submission 10115

---

> > > ### Author Response · Authors · 2024-12-03
> > > **Official Comment by Authors**
> > >
> > > Dear Reviewer mLed,
> > >
> > > Many thanks for your reply. It is glad to hear that your concerns have been addressed.
> > >
> > > If you feel that your concerns have been well-addressed, we do hope for an updated rating, as it would be a strong encouragement for our efforts.
> > >
> > > We noticed that the current score remains "borderline reject," despite **addressing all your concerns and no weaknesses being mentioned**. Could you kindly clarify **what prevents you from assigning a more positive score?** Your additional insights would be invaluable for us to further refine our work. There is still time to discuss, we are delighted to answer any questions or address any concerns from you.
> > >
> > > We are looking forward to your reply!
> > >
> > > Best regards,
> > >
> > > Authors of Submission 10115

---

### Official Review · Reviewer_LRdc · 2024-11-04

**Soundness:** 2
**Presentation:** 2
**Contribution:** 2
**Rating:** 5
**Confidence:** 3

**Summary:**

This paper proposes a new framework, called Sample-specific Score-aware Noise Injection (SSNI). SSNI uses a pre-trained score network to estimate how much a data point deviates from the clean data distribution, then adaptively adjusts the injected noise level for each sample, achieving sample-specific noise injections for diffusion-based purification.

**Strengths:**

A new framework that adaptively adjusts the injected noise level for each sample, thus achieving sample-specific noise injections for diffusion-based purification.

**Weaknesses:**

There are some unclear points that need more clarification; and insufficient theoretical justifications; see below for detailed questions.

**Questions:**

How is the standard accuracy calculated? Are the clean samples also purified before applying the pre-trained classifier network?

What is the value of t within s_\theta in eq (5)?

It is a bit confusing to me how the theoretical results in sec 3 support the claimed relationship between score norm and the optimal noise level. It seems like that Lemma 3 and Prop 1 are about the score norm of $p_t(x)$ where $p_t$ is the marginal distribution of $x_t$, generated by injecting sequential Gaussian noise to clean data $x_0$. In practice, the data may be attacked in different ways and contained by non-Gaussian noise. I think it would be more beneficial to present the theoretical insights for the score norm evaluated on contaminated data under certain attacks (maybe measured by the contamination level $\epsilon$), and how this norm depends on the contamination level $\epsilon$, thus showing that it will be significantly larger than the score norm of clean data.

---

> ### Author Response · Authors · 2024-11-22
> **Response to Reviewer LRdc**
>
> Thank you so much for your comments! Your thorough review and comments are very important to the improvement of our work! **Please find our replies below.**
>
> >Q1. How is the standard accuracy calculated?
>
> **Reply:** Thanks for your queston! **Standard accuracy** refers to the classification accuracy of a model on clean (non-adversarial) test samples after undergoing the purification process. In our work, when a natural image is input into the defense system, the classifier's accuracy is referred to as the **standard accuracy**. The error margin following the accuracy values in each table is calculated based on testing with three different random seeds. Our experimental results demonstrate that DBP methods, through our proposed framework, achieve a better trade-off between **standard accuracy** and **robust accuracy**.
>
> >Q2. Are the clean samples also purified before applying the pre-trained classifier network?
>
> **Reply:** **Yes.** This is mainly because **we cannot assume what type of test data we have**. In practice, the input data could either be **adversarial examples** or **clean examples**. Therefore, clean examples will also be purified before applying the pre-trained classifier network.
>
> >Q3. What is the value of t within s_\theta in eq (5)?
>
> **Reply:** Thanks for your question! This $t$ is a noise level specified in [1], $t \in (0, T)$ in the following equation: $\text{EPS}(\mathbf{x}) = \mathbb{E} _{t \sim \mathcal{U}(0, T)} \nabla _{\mathbf{x}} \log p _t(\mathbf{x}).$ We use the same settings as [1] specified for both CIFAR-10 and ImageNet-1K, where $t=20$ for CIFAR-10 and $t=50$ for ImageNet-1K.
>
> [1] Detecting Adversarial Data by Probing Multiple Perturbations Using Expected Perturbation Score, ICML 2023.
>
> >Q4. It is a bit confusing to me how the theoretical results in sec 3 support the claimed relationship between score norm and the optimal noise level. It seems that Lemma 3 and Prop 1 are about the score norm of pt(x) where pt is the marginal distribution of xt, generated by injecting sequential Gaussian noise to clean data x0. In practice, the data may be attacked in different ways and contained by non-Gaussian noise. I think it would be more beneficial to present the theoretical insights for the score norm evaluated on contaminated data under certain attacks (maybe measured by the contamination level ϵ), and how this norm depends on the contamination level ϵ, thus showing that it will be significantly larger than the score norm of clean data.
>
> **Reply:**  Thanks for your comment! However, there might be **some misunderstanding**. In Lemma 3 and Prop1, injecting Gaussian noise refers to **the forward process of diffusion models.** This process is **independent** of how the adversarial examples are generated. Lemma 3 and Prop 1 aim to motivate that the score norm can serve as a quantitative criterion for estimating the optimal noise level $t^*$ specific to each sample. Empirically, we demonstrate that **the cleaner (i.e., lower $\epsilon$) a sample is, the smaller its score norm tends to be**, which establishes a connection between the score norm and the maximum perturbation budget $\epsilon$. For more details, please kindly check Figure 4 in Appendix B in our manuscript.

---

> ### Author Response · Authors · 2024-11-22
> **Response to Reviewer LRdc**
>
> **In the end,** we want to thank you for providing this valuable feedback to us, which is always important to hear opinions from other experts. If you feel there are still unclear points regarding our paper, **please discuss** with us in the author-reviewer discussion phase. If you feel that your concerns are well-addressed, we do hope for an updated rating and new comments (if possible). Thanks again!

---

> ### Author Response · Authors · 2024-11-23
> **Reminder - Discussion Stage Closing Soon - 23 November**
>
> Dear Reviewer LRdc,
>
> We appreciate the time and effort that you have dedicated to reviewing our manuscript.
>
> We have carefully addressed all your queries. Could you kindly spare a moment to review our responses?
>
> Have our responses addressed your major concerns?
>
> If there is anything unclear, we will address it further. We look forward to your feedback.
>
> Best regards,
>
> Authors of Submission 10115

---

> ### Author Response · Authors · 2024-11-24
> **Reminder - Discussion Stage Closing Soon - 24 November**
>
> Dear Reviewer LRdc,
>
> We appreciate the time and effort that you have dedicated to reviewing our manuscript.
>
> We have carefully addressed all your queries. Could you kindly spare a moment to review our responses?
>
> Have our responses addressed your major concerns?
>
> If there is anything unclear, we will address it further. We look forward to your feedback.
>
> Best regards,
>
> Authors of Submission 10115

---

> ### Author Response · Authors · 2024-11-25
> **Reminder - Discussion Stage Closing Soon - 25 November**
>
> Dear Reviewer LRdc,
>
> We appreciate the time and effort that you have dedicated to reviewing our manuscript.
>
> We have carefully addressed all your queries. Could you kindly spare a moment to review our responses?
>
> Have our responses addressed your major concerns?
>
> If there is anything unclear, we will address it further. We look forward to your feedback.
>
> Best regards,
>
> Authors of Submission 10115

---

> ### Author Response · Authors · 2024-11-26
> **Reminder - Discussion Stage Closing Soon - 26 November**
>
> Dear Reviewer LRdc,
>
> We appreciate the time and effort that you have dedicated to reviewing our manuscript.
>
> We have carefully addressed all your queries. Could you kindly spare a moment to review our responses?
>
> Have our responses addressed your major concerns?
>
> If there is anything unclear, we will address it further. We look forward to your feedback.
>
> Best regards,
>
> Authors of Submission 10115

---

> ### Author Response · Authors · 2024-11-27
> **Reminder - Discussion Stage Closing Soon - 27 November**
>
> Dear Reviewer LRdc,
>
> We appreciate the time and effort that you have dedicated to reviewing our manuscript.
>
> We have carefully addressed all your queries. Could you kindly spare a moment to review our responses?
>
> Have our responses addressed your major concerns?
>
> If there is anything unclear, we will address it further. We look forward to your feedback.
>
> Best regards,
>
> Authors of Submission 10115

---

> > ### Comment · Reviewer_LRdc · 2024-11-27
> >
> > I thank the authors for their detailed explanation, which has addressed part of my previous questions about clean and robust accuracy, and I am sorry for my late reply.
> >
> > I will maintain my original score, and state some of my further and final comments below.
> >
> > According to the authors' clarification, if I understand it correctly, the $t$ within $s_\theta(x,t)$ in eq (5) should be $t^*$, a pre-specified noise level (which is the same as $T$ in alg 1). The authors may want to revise this to avoid further confusion. This is a very minor point.
> >
> > I maintain my original opinion about the lack of certain aspects of the presented theoretical results. I completely understand that $p_t$ refers to the marginal distribution of $x_t$ in the forward process after injecting Gaussian noise (and this is independent of adversarial attacks), and $s_\theta(x,t)$ refers to the score of this distribution $p_t$. However, I do not see a sufficient connection between the observation in Proposition 1 and the claim that "score norm can serve as a quantitative criterion for estimating the optimal noise level specific to each sample":
> >
> > From Prop 1, for a given sample $x$, the score norm of $s_\theta(x, t_1)$ and norm of $s_\theta(x, t_2)$ should be different for different $t_1,t_2$. However, in the algorithm, the time step $t^*$ (or $T$ in algo 1) is a fixed constant, i.e., the optimal noise level is selected based on the norm of $s_\theta(x, t^*)$. In other words, Prop 1 addresses differences in score norms across different time steps: "norms of $s_\theta(x, t_1)$ and $s_\theta(x, t_2)$ are different)"; while algo 1 is motivated by "norms of $s_\theta(x_1, t^*)$ and $s_\theta(x_2, t^*)$ should be different if the level of adversarial perturbation $\epsilon$ (and this attack is arbitrary, non-Gaussian, different from the forward process) is different within $x_1$ and $x_2$". This inconsistency weakens the theoretical contribution of the paper.
> >
> > I do appreciate the empirical results provided in Figure 4 in Appendix B, and I agree this can serve as empirical evidence to justify the choice of reweighting function and the over framework proposed in this paper. However, these empirical results alone do not significantly strengthen the theoretical contributions of the paper. After reviewing the discussion between the authors and other reviewers, I believe the paper would benefit from substantial revisions, such as additional content/explanations and maybe a reorganization/restatement of theoretical findings. Therefore, I maintain my original evaluation.

---

> > > ### Author Response · Authors · 2024-11-30
> > > **Response to Reviewer LRdc**
> > >
> > > Thank you very much for your follow-up questions.
> > > We appreciate the opportunity to **clarify our work's contributions.**
> > > This response will address two aspects as below.
> > >
> > > > Q1: Revise the notation of $s _{\theta}(x, t)$ in eq (5) with respect to $t$.
> > >
> > > Thank you for highlighting the ambiguity in the notation.
> > > We note that Algo 1 describes the SSNI framework for input test samples $\mathbf{x} = \{ \mathbf{x} _i \} _{i=1}^{n}$, representing $n$ different images, and $t _i$ used to compute the score is not necessarily the same across $n$ samples, i.e., we allow $t _i \neq t _j$, for $i \neq j$.
> > >
> > > Accordingly, we have revised Eq (5) and the accompanying explanation to reflect that noise levels $\{ t _i \} _{i=1}^n$ **can differ** across samples $\{\mathbf{x} _i \} _{i=1}^n$, **rather than** a single sample-shared $t^{*}$. Specifically:
> > > - Clarified Notation: The dependency of $s _\theta(\mathbf{x})$ on $t$ has been omitted to avoid confusion. Instead, the revised text explicitly states that $s _\theta(\mathbf{x})$ is computed using noise levels based on the characteristics of each sample (in Section 4.1).
> > > - Appended Explanation: **Section 4.2 now discusses the practical difficulty of determining $t _i$ for individual samples and justifies our use of EPS [1] for robust score estimation.**
> > >
> > > ---
> > >
> > > > Q2: Reorganize Section 3 to clarify the connection between Prop 1, Figure 4 and motivation of SSNI
> > >
> > > We have completely **reorganized** Section 3 to sort out the
> > > connection between the observations (Figure 1 and 4), Prop 1, and the **overall motivation** of propsing the SSNI framework, as well as addressing **potential misunderstandings** of our manuscript.
> > > Section 3 now presents the observations in a logical progression below.
> > >
> > > **Observation 1: Connection between $t^{*}$ and $\epsilon$**. From Figure 1, we empirically observe that using a fixed sample-shared noise level $t^*$ leads to suboptimal purification outcomes for adversarial examples with varying perturbation budgets $\epsilon$ - a *need for sample-specific noise injection levels tailored to individual $\epsilon$* - how to determine the noise level for different samples then?
> > >
> > > **Observation 2: Connection between $\epsilon$ and score norms**.
> > > As adversarial samples are considered OOD to clean examples [2], which can be detected through score norms [3], we investigate how different $\epsilon$ affects score norms of adversarial examples.
> > > Figure 4 shows that higher $\epsilon$ tend to lead to higher score norms and vince versa.
> > > Naturally, score norms can differentiate adversarial examples based on their perturbation strength, supporting the use of score norms for guiding noise selection.
> > >
> > > **Observation 3: Connection between score norms and $t^{*}$**.
> > > Since in diffusion-based purification, test samples (regardless of clean or adversarial) need to be purified by traversing the forward/reverse diffusion processes, we leverage the property of diffusion models, and find that different noise levels correlate with different score norms in Prop 1.
> > > Importantly, Prop 1 applies to **different samples** along the diffusion chain, **not to the same sample** in different $t$.
> > > This motivates our **hypothesis** that score norms could guide the selection of sample-specific noise levels $t _i$.
> > >
> > > **These three observations collectively motivate SSNI**.
> > > The score norms reflect the "**denoising effort**" required for each sample regardless of clean or adversarial examples, and correlate with the adversarial perturbation strength of each sample.
> > > By leveraging this connection, SSNI dynamically adjusts noise injection levels based on each sample's score norm, aligning the purification (denoising) process with the specific perturbation budget.
> > >
> > >
> > > Importantly, we **do not want to overclaim** the theoretical contribution of Prop 1.
> > > In Remark 1 of Prop 1, we **have acknowledged the simplification** made to present Prop 1:
> > > - We focused on the property of diffusion models, disregarded the potential interactions between Gaussian and adversarial noise (which is often difficult to analyze directly).
> > > - In this sense, Prop 1 **only** serves to connect different score norms to different noise levels $t^*$ **in the context of diffusion model**-based purification, **motivates** our adoption of score norms to reweight $t^*$, but not a comprehensive theorem and justification.
> > > - We note that the **SSNI framework itself is the primary contribution**, driven by empirical evidence in Section 5.
> > >
> > > We hope this reorganization and clarification enhance the manuscript’s readability and address your concerns.
> > > Thank you again for helping us improve our manuscript.
> > >
> > >
> > > [1] Detecting adversarial data by probing multiple perturbations using expected perturbation score. In ICML 2023
> > >
> > > [2] Maximum mean discrepancy test is aware of adversarial attacks. in ICML 2021
> > >
> > > [3] Adversarial purification with score-based generative models. in ICML 2021

---

> > > ### Author Response · Authors · 2024-11-30
> > > **Response to Reviewer LRdc**
> > >
> > > **In the end,** we want to thank you for providing this valuable feedback to us, which is always important to hear opinions from other experts in this field. If you feel there are still unclear points regarding our paper, **please discuss** with us in the author-reviewer discussion phase. If you feel that your concerns are well-addressed, we do hope for an updated rating and new comments (if possible). Thanks again!

---

> > > > ### Comment · Reviewer_LRdc · 2024-11-30
> > > >
> > > > Thank you for the additional response. After reviewing your reply and the revised manuscript, I have decided to maintain my original score.
> > > >
> > > > _Notation_
> > > > The notation has become more confusing after omitting the dependency of $s_\theta(x)$ on t. It remains unclear how this score is computed. Now the authors claim that the score in the reweighting function is computed using noise levels based on the characteristics of each sample (however these characteristics and the method for determining the noise level are not clearly discussed). It becomes more clear in Sec 4.3 when the authors mention the usage expected perturbation score (EPS) for computing score, it should be explicitly clarified earlier (and in Alg 1) if EPS is indeed the method implemented in the proposed algorithm. In the way it is presented now, it introduces more ambiguity for the score computation.
> > > >
> > > > _About Prop 1 and motivation
> > > > I do appreciate Fig 1 and 4 as motivations for the proposed method. However, prop1 has very limited relevance to these motivations. And if the authors mean "applies to different samples along the diffusion chain, not to the same sample in different t", then it probably should be $s(x_1,t_1)$ and $s(x_2,t_2)$, where $s(x,t)$ denotes the score function of marginal distribution $p_t$, and $x_1,x_2$ represent the data values at time $t_1,t_2$ starting at the same initial value (i.e., in the same diffusion chain).
> > > >
> > > > Overall and again, I appreciate the author's detailed response and effort in trying to clarify the contributions. Given the substantial amount of revisions authors have undertaken and also some remaining areas that can still be improved, I would like to maintain my score and I believe this work would benefit from a more thorough revision to fully clarify its contributions and enhance its readiness for future publication.

---

### Official Review · Reviewer_2MWZ · 2024-11-04

**Soundness:** 2
**Presentation:** 3
**Contribution:** 2
**Rating:** 6
**Confidence:** 2

**Summary:**

Neural networks are highly susceptible to adversarial noise, which compromises their robustness in critical applications. Diffusion-based purification (DBP) methods address this by introducing Gaussian noise in a controlled process to counteract adversarial noise, typically using a fixed noise level $t*$ across all samples. However, this uniform approach can lead to suboptimal results, as the ideal noise level varies by sample. In response, the authors propose Sample-specific Score-aware Noise Injection (SSNI), a framework that adaptively adjusts $t*$ based on a sample’s deviation from the clean data distribution, estimated through score norms. By tailoring noise levels—lower for cleaner samples and higher for noisier ones—SSNI enhances both accuracy and robustness across benchmarks like CIFAR-10 and ImageNet-1K, outperforming traditional DBP methods.

**Strengths:**

1. The need for sample-based noise injection is intuitive and well-motivated. I really like Figure 1.
2. The paper is well-written and easy to follow.

**Weaknesses:**

First, I would like to make a note that my background is more on diffusion rather than adversarial perturbation. Thus, I am not sure about the significance of the empirical performance of the proposed method, compared to the baselines. I will leave the assessment of this part to the other reviewers more qualified than me.

My biggest concern with the paper is that a lot of the arguments and designs feel hand-wavy without proper principles or justification. I will write the detailed point below.

1. I would like to point out that diffusion models are never trained to model the score directly, but scaled versions of its variants, and only represent scores through reparametrizations. Typically, the training objective of diffusion models is parameterized so that the model's prediction has a **constant** norm across noise levels (for the ease of training), for example, the $\epsilon$-prediction in DDPM and the more modern EDM [1,2]. The difference in terms of the score norm comes only from $t$ through reparametrization. Essentially, in terms of diffusion model training, they are not trained to differentiate the norm of the scores across noise levels. This property holds for the two models the authors used in experiments. Thus, I am wondering what exactly the authors observe here in Figure 4 in Appendix B. How is this score norm calculated? Is this calculated through a pre-trained model or through some analytic solution? If the pre-trained models' output norm really correlates with the sample $x$ (and we should keep the input of time/noise level constant), then it is really an undesired outcome of those models. Then what about the more modern, better-trained models, like EDM?
2. I am also confused by the logic that the further "away" the sample is from the "clean" distribution, the more noise we need to inject for purification. In particular, why are we adding a proportional amount of noise estimated in the sample again? For example, if the procedure estimates the model is at time $t$, Algorithm 1 essentially forwards the sample to $t+t'$ time right? I don't understand the logic behind this choice. Also, now when you run the backward sampling process, do you start at $t+t'$ or $t'$? Essentially, when you do the forward and backward, are you treating the test (start) sample $x$ as a clean or a noisy one in terms of diffusion?
3. The two realizations of the reweighting function have no justification at all. Also, how is the bias term $b$ selected?

In all, I feel the main insights of the paper are based on some misunderstanding of the current diffusion models. Also, a lot of the design choices feel questionable, and thus heuristic to me. In the end, I feel unconvinced by the paper's argument and proposal. However, I am happy to be corrected.

[1] Karras et al. Elucidating the Design Space of Diffusion-Based Generative Models. NeurIPS 2022.

[2] Karras et al. Analyzing and Improving the Training Dynamics of Diffusion Models. CVPR 2024.

**Questions:**

1. The sub-caption of Figure 1 can be simplified (shared).
2. Judging from Equation 7, are you suggesting to evaluate the pre-trained diffusion model on many $t$s? If so, this requires many network evaluations, right?
3. On Line 344, I think you mean "the key idea is to scale the coefficient" right? Instead of scaling $\|EPS(x)\|$.
4. The time overhead reported in Section 5.5 is not very meaningful. Some relative measurements are more desired.

---

> ### Author Response · Authors · 2024-11-22
> **Response to Reviewer 2MWZ (overall clarification - part 1)**
>
> Thank you so much for your comments! It is our pleasure that the motivation of our paper can be recognized. Your thorough review and comments are very important to the improvement of our work! **Please find our replies below.**
>
> > In all, I feel the main insights of the paper are based on some misunderstanding of the current diffusion models. Also, a lot of the design choices feel questionable, and thus heuristic to me. In the end, I feel unconvinced by the paper's argument and proposal. However, I am happy to be corrected.
>
>  **Reply:** Thanks for your comments! First of all, although our work uses diffusion models, the problem setting in our work **differs significantly** from the traditional generative tasks typically associated with diffusion models. Meanwhile, we realize that our manuscript lacks details of diffusion-based adversarial purification and background knowledge of adversarial perturbations, which might lead to **potential misunderstanding** of our work.
>
> Therefore, we would like to clarify the problem setting and the background knowledge of our work at the early stage of our response. To address this comprehensively, we have divided our response into several sections:
>
> **Section 1: the high-level objective of our work and some preliminaries**
>
> The high-level objective of our work is to remove adversarial perturbations from adversarial examples before feeding them into the underlying classifier for classification prediction.
>
> Thus, the task here is **image classification**, instead of image generation. **Diffusion model serves as a tool to remove adversarial perturbations.** Then, we will introduce some preliminaries to better understand our work.
>
> - **Adversarial examples (AEs)** are often crafted by adding imperceptible **adversarial perturbations** to clean examples (CEs), which can easily mislead a well-trained deep learning model to make wrong predictions. Usually, adversarial perturbations are obtained by **maximizing cross-entropy loss** and it is typically constrained within a specified norm (e.g., $\ell_p$​-norm) to ensure it **remains imperceptible**. Please note that **adversarial perturbations are significantly different from Gaussian noise**.
> - **Diffusion-based purification (DBP)** has recently emerged as **a prominent solution** to remove the adversarial perturbations from AEs, **leveraging the denoising capabilities of diffusion models.** DiffPure [1] is a well-known framework in DBP, which uses diffusion models as purifiers to remove the adversarial perturbations. The philosophy behind this is that adding Gaussian noise to the AEs during the forward process of diffusion models can align the distribution of AEs towards a Gaussian distribution. Then, by iteratively denoising the noise-injected AEs at $t^*$ using a reverse diffusion process, the adversarial perturbations can be effectively 'washed out'.
>
> [1] Diffusion Models for Adversarial Purification, ICML 2022.
>
> **Section 2: key differences between the role of diffusion models in DBP and image generations**
>
> Here, we summarize some **key differences** between the role of diffusion models in DBP and image generations
> || **Diffusion Models in Image Generation**|**Diffusion Models in Adversarial Purification**|
> |--|--|--|
> | **Objective**| Generate high-quality samples (e.g., images, audio, or video).| Remove adversarial perturbations from adversarial examples and restore clean examples.|
> | **Input**| Random noise as the starting point of the generation process.|Adversarial examples / clean examples. *Note that the goal is to purify adversarial examples, but we cannot assume the type of input data at the inference stage. Therefore, clean examples will also go through the diffusion model for purification.*|
> | **Output**| Newly generated samples without direct clean references.| Purified samples, closely resembling their original clean state.|
> | **Utilization of Process**| Full reverse diffusion process to transform noise into samples.| Denoising tool to remove adversarial perturbations, not for full generation.|

---

> ### Author Response · Authors · 2024-11-22
> **Response to Reviewer 2MWZ (overall clarification - part 2)**
>
> **Section 3: research gap in DBP and our motivation**
>
> To use diffusion models to remove adversarial perturbations, **a natural challenge in this field** is how much Gaussian noise should be injected into the input sample during the forward process of diffusion models [1], which is a **trade-off problem**:
> - If the Gaussian noise is not enough, then adversarial perturbations cannot be fully removed after the reverse process.
> - If the Gaussian noise is too much, then the purified samples will lose their original semantic meanings (i.e., in this case, it is more like generating new images instead of recovering original images).
>
> In practice, the amount of Gaussian noise injected to the samples is controlled by the timestep $t^*$, and existing DBP methods manually select a constant $t^*$ for all samples.  However, in this paper, we find that using a sample-shared $t^*$ may overlook the fact that an optimal $t^*$ indeed could be different at sample-level. For example, some adversarial examples only require a small $t^*$ to be recovered. Therefore, a sample-shared $t^*$ will destroy the semantic meaning of these examples, affecting the classification performance. Meanwhile, some adversarial examples require a large $t^*$ to be recovered. Therefore, a sample-shared $t^*$ cannot effectively remove the adversarial noise.
>
> [1] Diffusion Models for Adversarial Purification, ICML 2022.
>
> **Section 4: contributions of our work**
> - We empirically observe that using a sample-shared $t^*$ may overlook the fact that an optimal $t^*$ could indeed be different at sample-level. This could be used as a strong motivation to address the above-mentioned challenges.
> - Based on this motivation, we further propose Sample-specific Score-aware Noise Injection (SSNI), a new framework that leverages score norms estimated by a pre-trained score network to adaptively adjust $t^*$ on a sample-specific basis.
> - Extensive experiments show that SSNI can boost clean accuracy and robust accuracy simultaneously by a notable margin against well-designed adaptive white-box attacks.
>
> In general, the success of SSNI takes root in the following aspects: (1) an optimal noise level $t^*$ for each sample indeed could be different, making SSNI a more effective approach to unleash the intrinsic strength of DBP methods; (2) existing DBP methods often inject excessive noise into clean samples, resulting in a degradation in clean accuracy. By contrast, SSNI injects less noise into clean samples, and thereby notably improving the clean accuracy. Meanwhile, SSNI can effectively handle adversarial samples by injecting sufficient noise on each sample; (3) SSNI is designed as a general framework instead of a specific method, allowing it to be seamlessly integrated with a variety of existing DBP methods.

---

> ### Author Response · Authors · 2024-11-22
> **Response to Reviewer 2MWZ (Weakness 1)**
>
> >W1.1 I would like to point out that diffusion models are never trained to model the score directly ...Typically, the training objective of diffusion models is parameterized so that the model's prediction has a constant norm across noise levels (for the ease of training) ... Essentially, in terms of diffusion model training, they are not trained to differentiate the norm of the scores across noise levels. This property holds for the two mode ls the authors used in experiments. Thus, I am wondering what exactly the authors observe here in Figure 4 in Appendix B.
>
> **Reply:**  Thanks for your comments! We **agree** with what you describe about the training objective of diffusion models. However, in our paper, **the score $\nabla_\mathbf{x} \log p_t(\mathbf{x})$ can be estimated through the score network $s_\theta(\mathbf{x}_t, t)$. that has a training objective of score matching [1],** which is indeed different from what you mentioned in the question. The training objective minimizes:
>
> $$\frac{1}{2} \mathbb{E} _{q _\sigma(\tilde{\mathbf{x}} \mid \mathbf{x}) p _{\text{data}}(\mathbf{x})}
> \left[ \| \mathbf{s} _\theta(\tilde{\mathbf{x}}) - \nabla _{\tilde{\mathbf{x}}} \log q _\sigma(\tilde{\mathbf{x}} \mid \mathbf{x}) \| _2^2 \right].$$
>
> Therefore, we believe there is **a misunderstanding** due to a lack of details regarding how the score norm is estimated. We will clarify this part in the revised version of our manuscript.
>
> Intuitively, we can **consider adversarial examples as a special case of out-of-distribution (OOD) data** relative to the original data distribution. Therefore, the score $\nabla _\mathbf{x} \log p _t(\mathbf{x})$ of adversarial examples will differ from the score of the clean data. As supported by other studies, **adversarial examples tend to have larger score norms** [2] [3].
>
> We further validate this phenomenon in Figure 4 by showing that **with the increase of attack strength** (i.e., the larger $\epsilon$ means the example contains more adversarial perturbations), **the score norms will also increase.** Hence, this could explain that data further away from the original distribution would have greater momentum (score norm) towards the high data intensity area in vector field [1].
>
> [1] Generative Modeling by Estimating Gradients of The Data Distribution, NeurIPS 2019.
>
> [2] Adversarial Purification with Score-Based Generative Models, ICML 2021.
>
> [3] Robust Evaluation of Diffusion-Based Adversarial Purification, ICCV 2023.
>
> > W1.2 How is this score norm calculated? Is this calculated through a pre-trained model or through some analytic solution? If the pre-trained models' output norm really correlates with the sample x (and we should keep the input of time/noise level constant), then it is really an undesired outcome of those models.
>
> **Reply:** Thanks for your question and sorry for the confusion! Before moving on to how to calculate EPS, we would like to clarify the calculation of score $\nabla _\mathbf{x} \log p _t(\mathbf{x})$ in the early stage.
>
> Specifically, the score $\nabla_\mathbf{x} \log p_t(\mathbf{x})$ **does not have the analytic solution**. Therefore, the score is estimated by training a neural network with the objective of score matching [2]. As a result, the score $\nabla_\mathbf{x} \log p_t(\mathbf{x})$ can be estimated through the score network $s_\theta(\mathbf{x}_t, t)$. In practice, we use a pre-trained score network in score-sde to achieve the estimation of the score $s _\theta(\mathbf{x} _t, t)$ [3].
>
> The score in Figure 4 refers to the **Expected Perturbation Score (EPS)**, which is computed from the score function in the Score SDE. The detailed implementation of EPS calculation can be found here: [code](https://github.com/ZSHsh98/EPS-AD/blob/cceb45db9c97ccec255b8ec18750c6e36acbf6f1/eval_epsad.py#L123) [1]. Here, we first select a range $(0, T)$, and then use each perturbed data $\mathbf{x_t}$, $t \in (0, T)$ to input to the score network and it directly returns the corresponding score of $\mathbf{x_t}$. Hence, the EPS is computed based on the expectation of these scores. This does not require any network evaluations. For CIFAR-10, we use $T = 20$ and for ImageNet, we use $T = 50$, which follows the same setting mentioned in [1].
>
> [1] Detecting Adversarial Data by Probing Multiple Perturbations Using Expected Perturbation Score, ICML 2023.
>
> [2] Generative Modeling by Estimating Gradients of The Data Distribution, NeurIPS 2019.
>
> [3] Score-Based Generative Modeling Through Stochastic Differential Equations, ICLR 2021.
>
> >W1.3 Then what about the more modern, better-trained models, like EDM?
>
> **Reply:** Indeed, the score network in EDM framework is **same** as what we used to estimate the score (i.e., both of them are based on the Score SDE) [1]. Therefore, the performance will not be changed.
>
> [1] Elucidating the Design Space of Diffusion-Based Generative Models, NeurIPS 2022.

---

> ### Author Response · Authors · 2024-11-22
> **Response to Reviewer 2MWZ (Weakness 2)**
>
> >W2.1 I am also confused by the logic that the further "away" the sample is from the "clean" distribution, the more noise we need to inject for purification. In particular, why are we adding a proportional amount of noise estimated in the sample again?
>
> **Reply:** Thanks for your question and sorry for the confusion!
> - Firstly, **the "sample" in the sentence "the further away the sample is" refers to adversarial examples**, instead of the Gaussian noise-injected samples after the forward process.
> - Secondly, the "noise" in the sentence "the more noise we need to inject for purification" here refers to **the Gaussian noise added in the forward diffusion process**.
>
> **The logic/philosophy behind it is:** Adding Gaussian noise to adversarial examples is widely recognized as an effective defense method in diffusion-based purification. In DiffPure, for instance, the theoretical foundation suggests that both adversarial and clean samples are aligned towards a Gaussian distribution when Gaussian noise is injected. This point is also discussed in a recent work [1]. The purpose of injecting Gaussian noise during the forward process is to effectively "cover" the adversarial perturbations, and thus they can be 'washed out' through the reverse process.
>
> [1] Towards Understanding The Robustness of Diffusion-Based Purification: A Stochastic Perspective, arxiv 2024.
>
> [2] Diffusion Models for Adversarial Purification, ICML 2022.
>
> >W2.2 For example, if the procedure estimates the model is at time t, Algorithm 1 essentially forwards the sample to t+t’ time right? I don't understand the logic behind this choice.
>
> **Reply:** Thanks for your question! However, there might be **some misunderstanding**. Indeed, we do not introduce an additional $t'$. The detailed explanation of Algorithm 1 is as follows:
> - Given an input $\mathbf{x}$ (e.g., it could be an adversarial example or clean example), we use a pretrained score network to calculate the score norm of $\mathbf{x}$.
> - Then we feed the score norm into a reweighting function, which outputs $t(\mathbf{x})$.  This $t(\mathbf{x})$ serves as the sample-specifc $t^*$s for purifying the sample $\mathbf{x}$.
>
> Intuitively, we are making the optimal timestep $t^*$ adaptive for each sample based on its score norm.
>
> >W2.3 Also, now when you run the backward sampling process, do you start at t+t’ or t’ ?
>
> **Reply:** Thanks for your question! Our proposed framework assigns an independent timestep $t(\mathbf{x})$ to each sample image (i.e., sample-specific $t^*$ for each sample). Therefore, each image starts the backward sampling process from a sample-specific $t^*$ that **can be different** from other images.
>
> >W2.4 Essentially, when you do the forward and backward, are you treating the test (start) sample x as a clean or a noisy one in terms of diffusion?
>
> **Reply:** Thanks for your question! In terms of diffusion, every input begins as the initial image (i.e., $x_0$ at t = 0). Different from the image generation task, here the input is not random noise. Instead, $x_0$ here could **either be a clean example or an adversarial example.** This is because we cannot assume the type of test data we have beforehand.  Also, it is important to note that the noise in adversarial examples (i.e., adversarial perturbations) is **fundamentally different** from the Gaussian noise used in the diffusion process.

---

> ### Author Response · Authors · 2024-11-22
> **Response to Reviewer 2MWZ (Weakness 3)**
>
> >W3. The two realizations of the reweighting function have no justification at all. Also, how is the bias term b selected?
>
> **Reply:**
> - **Justification:** The design principle of the reweighting function is to scale the timestep $t^*$ based on the score information. As we have observed that clean data often has a much lower norm value than adversarial one, we believe that incorporating a linear function or a sigmoid function would reweight the timestep properly. For realizations, we extract 5,000 validation clean examples from the training data (denoted as  $\mathbf{x}_v$) and we use $\left\|\text{EPS}(\mathbf{x}_v)\right\|$ as a reference to indicate the approximate EPS norm values of clean data, which can help us reweight $t^*$. Thus, based on the EPS norm information, the reweighting function assigns adaptive timestep accordingly.
> - **How to select bias term b:** When we select the bias term, we first split a validation set from each dataset and record the performance in above tables. We then choose the value based on the validation performance, then we use the term in our test data evaluations.
>
> We apologize that our manuscript lacks of ablation study of the bias term. We present the result below. The bias term has the same magnitude with the timestep $t^*$. Also, there is a small mistake in Equation 9, the correct formula should be:
> $f_\sigma(\left\|\text{EPS}(\mathbf{x})\right\|,~t^*) = \frac{t^* + b}{1 + \exp\{-(\left\|\text{EPS}(\mathbf{x})\right\|-\mu)/\tau\}}$
>
> Table 1: Ablation study on the bias term $b$. We report clean and robust accuracy (%) against white-box PGD+EOT adaptive attacks ($\ell_\infty, \epsilon=8/255$) on CIFAR-10. The classifier used is WRN-28-10.
> |Classifier|Bias|Standard|Robust|
> |--|--|--|--|
> |WRN-28-10|0|**94.34$\pm$ 1.43**|55.27$\pm$ 0.75|
> |WRN-28-10|5|93.95$\pm$ 1.17|57.23$\pm$ 1.62|
> |WRN-28-10|10|93.17$\pm$ 1.05|57.03$\pm$ 0.54|
> |WRN-28-10|15|92.38$\pm$ 1.29|57.64$\pm$ 0.89|
> |WRN-28-10|20|92.38$\pm$ 1.29|57.64$\pm$ 0.89|
> |WRN-28-10|25|92.10$\pm$ 1.03|58.24$\pm$ 1.22|
> |WRN-28-10|30|92.97$\pm$ 0.37|**59.18$\pm$ 1.65**|
>
>
> Table 2: Ablation study on the bias term $b$. We report clean and robust accuracy (%) against white-box PGD+EOT adaptive attacks ($\ell_\infty, \epsilon=8/255$) on CIFAR-10. The classifier used is WRN-70-16.
> |Classifier|Bias|Standard|Robust|
> |--|--|--|--|
> |WRN-70-16|0|94.34$\pm$ 0.45|56.45$\pm$ 1.22|
> |WRN-70-16|5|93.82$\pm$ 1.56|57.03$\pm$ 0.78|
> |WRN-70-16|10|**94.73$\pm$ 1.34**|58.10$\pm$ 0.24|
> |WRN-70-16|15|92.97$\pm$ 0.89|58.79$\pm$ 1.48|
> |WRN-70-16|20|92.97$\pm$ 0.89|**59.57$\pm$ 1.19**|
> |WRN-70-16|25|92.58$\pm$ 0.67|58.59$\pm$ 0.52|
> |WRN-70-16|30|92.77$\pm$ 0.94|58.59$\pm$ 0.52|
>
>
> Table 3: Ablation study on the reweighting function parameter $b$. We report clean and robust accuracy (%) against white-box PGD+EOT adaptive attacks ($\ell_\infty, \epsilon=4/255$) on ImageNet-1K. The classifier used is ResNet-50.
> |Classifier|Bias|Standard|Robust|
> |--|--|--|--|
> |ResNet-50|0|71.68$\pm$ 1.12|39.93$\pm$ 0.34|
> |ResNet-50|25|71.73$\pm$ 1.49|40.28$\pm$ 0.28|
> |ResNet-50|50|**71.96$\pm$ 0.13**|**43.88$\pm$ 0.22**|
> |ResNet-50|75|68.80$\pm$ 0.74|41.45$\pm$ 0.38|
> |ResNet-50|100|68.41$\pm$ 0.59|43.02$\pm$ 0.41|
> |ResNet-50|125|67.63$\pm$ 1.08|40.87$\pm$ 0.92|
> |ResNet-50|150|66.45$\pm$ 1.53|40.05$\pm$ 0.67|
>
> From the experimental results, we can derive the following conclusions: the selection of bias term does not impact the performance of our framework under CIFAR-10 and ImageNet-1K. Note that when bias increases, there is a general observation that the clean accuracy drops and the robust accuracy increases. This perfectly aligns with the understanding of optimal timestep $t^*$ selection in the previous DBP work, where large timestep would lead to drop of both clean and robust accuracy and small timestep cannot remove the adversarial perturbation effectively (see Figure 1 in [1]).
>
> [1] Robust Evaluation of Diffusion-Based Adversarial Purification, ICCV 2023.

---

> ### Author Response · Authors · 2024-11-22
> **Response to Reviewer 2MWZ (Questions)**
>
> >Q1. The sub-caption of Figure 1 can be simplified (shared)
>
> **Reply:** Thanks for your suggestion! We **will** simplify the sub-caption of Figure 1 in the revised version of our manuscript!
>
> >Q2. Judging from Equation 7, are you suggesting to evaluate the pre-trained diffusion model on many ts? If so, this requires many network evaluations, right?
>
> **Reply:** Thanks for your question! Indeed, **this process does not require many network evaluations**. With a reference to Eq.7 in our manuscript:  $\text{EPS}(\mathbf{x}) = \mathbb{E} _{t \sim \mathcal{U}(0, T)} \nabla _{\mathbf{x}} \log p _t(\mathbf{x}).$
> We have previously discussed how to estimate the score function $\nabla _{\mathbf{x}} \log p _t(\mathbf{x})$ through a score network $s _\theta(\mathbf{x _t}, t)$. Here, we first select a range $(0, T)$, and then use each perturbed data $\mathbf{x _t}$, $t \in (0, T)$ to input to the score network and it directly returns the corresponding score of $\mathbf{x _t}$. Hence, the EPS is computed by the expectation over these scores. This does not require many network evaluations. For CIFAR-10, we use $T = 20$ and for ImageNet, we use $T = 50$, which follows the same setting mentioned in [1].
>
> [1] Detecting Adversarial Data by Probing Multiple Perturbations Using Expected Perturbation Score, ICML 2023.
>
> >Q3. On Line 344, I think you mean "the key idea is to scale the coefficient" right? Instead of scaling |EPS(x)|.
>
> **Reply:** Thanks for your question and sorry for the confusion!  There might be a **potential misunderstanding** here. We believe scaling $∥\text{EPS}(\mathbf{x})∥$ is **more appropriate** here. Specifically, the term $\frac{\left\|\text{EPS}(\mathbf{x})\right\| - \xi _{\min}}{\xi _{\max} - \xi _{\min}}$ is explicitly designed to **normalize** $∥\text{EPS}(\mathbf{x})∥$ to a fixed range (typically 0-1) based on the minimum and maximum $∥\text{EPS}(\mathbf{x})∥$ values.  Therefore, the goal here is to scale $∥\text{EPS}(\mathbf{x})∥$ so that the coefficient of $t^*$ (i.e., $\frac{\left\|\text{EPS}(\mathbf{x})\right\| - \xi _{\min}}{\xi _{\max} - \xi _{\min}}$) is within the range of 0-1.  To avoid confusion, we will clarify this sentence in the revised version of our manuscript!
>
> >Q4. The time overhead reported in Section 5.5 is not very meaningful. Some relative measurements are more desired.
>
> **Reply:** Thanks for your comment! We will clarify the reasons why we report the inference time in our paper. DBP methods are **inference-time adversarial defense methods** (i.e., this kind of approach does not require any training process as it uses pre-trained diffusion models and pre-trained classifiers). Then, the consequence is that **the inference time for DBP methods is very slow**, which is a general limitation in this field. Given this limitation, our method further adds an additional module (i.e., the reweighting module) into the DBP framework, which will **inevitably increase the inference time**. Then, a natural question is: **''whether the increase in the inference time is affordable and acceptable given the improvements in the classification performance?"** The answer to this question is *affirmative*. Compared to the DBP baseline methods, this reweighting process is relatively *lightweight*, ensuring that SSNI is computationally feasible and can be applied in practice with minimal overhead.

---

> ### Author Response · Authors · 2024-11-22
> **Response to Reviewer 2MWZ**
>
> **In the end,** we want to thank you for providing this valuable feedback to us, which is always important to hear opinions from other experts. If you feel there are still unclear points regarding our paper, **please discuss** with us in the author-reviewer discussion phase. If you feel that your concerns are well-addressed, we do hope for an updated rating and new comments (if possible). Thanks again!

---

> ### Author Response · Authors · 2024-11-23
> **Reminder - Discussion Stage Closing Soon - 23 November**
>
> Dear Reviewer 2MWZ,
>
> We appreciate the time and effort that you have dedicated to reviewing our manuscript.
>
> We have carefully addressed all your queries. Could you kindly spare a moment to review our responses?
>
> Have our responses addressed your major concerns?
>
> If there is anything unclear, we will address it further. We look forward to your feedback.
>
> Best regards,
>
> Authors of Submission 10115

---

> ### Author Response · Authors · 2024-11-24
> **Reminder - Discussion Stage Closing Soon - 24 November**
>
> Dear Reviewer 2MWZ,
>
> We appreciate the time and effort that you have dedicated to reviewing our manuscript.
>
> We have carefully addressed all your queries. Could you kindly spare a moment to review our responses?
>
> Have our responses addressed your major concerns?
>
> If there is anything unclear, we will address it further. We look forward to your feedback.
>
> Best regards,
>
> Authors of Submission 10115

---

> ### Author Response · Authors · 2024-11-25
> **Reminder - Discussion Stage Closing Soon - 25 November**
>
> Dear Reviewer 2MWZ,
>
> We appreciate the time and effort that you have dedicated to reviewing our manuscript.
>
> We have carefully addressed all your queries. Could you kindly spare a moment to review our responses?
>
> Have our responses addressed your major concerns?
>
> If there is anything unclear, we will address it further. We look forward to your feedback.
>
> Best regards,
>
> Authors of Submission 10115

---

> ### Author Response · Authors · 2024-11-26
> **Reminder - Discussion Stage Closing Soon - 26 November**
>
> Dear Reviewer 2MWZ,
>
> We appreciate the time and effort that you have dedicated to reviewing our manuscript.
>
> We have carefully addressed all your queries. Could you kindly spare a moment to review our responses?
>
> Have our responses addressed your major concerns?
>
> If there is anything unclear, we will address it further. We look forward to your feedback.
>
> Best regards,
>
> Authors of Submission 10115

---

> > ### Comment · Reviewer_2MWZ · 2024-11-26
> >
> > I thank the authors for their detailed explanation, especially their extended coverage of the related background. They are very educational.
> >
> > Regarding the score norm: I did not know the authors were actually using the unnormalized DSM objective-based model [1] to calculate the score norm. Then, yes, the model's output could contain some information regarding the norm of the score. Still, I must point out that nowadays, this particular objective is no longer used, even in their follow-up [2]. This is certainly the case for [3] (which the authors claimed to use in later parts of the response? I feel there is a contradiction here, so I ask the authors to really make it clear how exactly this is done), and [4]. However, it is clear that this is an established empirical observation [5], so I will not fight against its validity. I feel there might be something else at play, as [5] also used a normalized training objective. Again, I certainly feel that this part needs to be written more clearly, because I feel like even in the author's response there are some contradictions: e.g. is it using [1] or [3]? Do you add Gaussian noise to the (clean/adversarial) sample before making that calculation?
> >
> > Some random comments: you could calculate the score analytically for some small datasets (see [4] Fig1.b), which may be interesting to look into.
> >
> > In all, I feel like the authors have cleared many of my questions and the proposed method is an interesting one, so I will raise my score.
> >
> > [1] Generative Modeling by Estimating Gradients of The Data Distribution, NeurIPS 2019.
> >
> > [2] Improved techniques for training score-based generative models, NeurIPS 2020.
> >
> > [3] Score-Based Generative Modeling Through Stochastic Differential Equations, ICLR 2021.
> >
> > [4] Elucidating the Design Space of Diffusion-Based Generative Models. NeurIPS 2022.
> >
> > [5] Adversarial Purification with Score-Based Generative Models, ICML 2021.

---

> > > ### Author Response · Authors · 2024-11-26
> > > **Many thanks for your reply and increasing your score to 6!**
> > >
> > > Dear Reviewer 2MWZ,
> > >
> > > Many thanks for your valuable comments again!
> > >
> > > We are currently reviewing the new concerns you’ve raised and will address them comprehensively. We aim to provide a detailed response as soon as possible.
> > >
> > > We also want to share with you the current situation regarding the other reviews. Unfortunately, **Reviewer mLed gave a rejection recommendation despite acknowledging that we addressed all the concerns**, and **another reviewer (Reviewer LRdc) has not yet to provide any response at all.**
> > >
> > > This has placed our submission in a **particularly challenging position (i.e., the averaged rating is around borderline).** We would be deeply grateful for your consideration of further raising your confidence or score, if we can address your follow-up concerns to your satisfaction.
> > >
> > > Your support would play a crucial role in ensuring our paper can be fairly evaluated in this rebuttal.
> > >
> > > Best regards,
> > >
> > > Authors of Submission 10115

---

> > > ### Author Response · Authors · 2024-11-27
> > > **Response to Reviewer 2MWZ further concerns - (1/2)**
> > >
> > > >Q1. Regarding the score norm: I did not know the authors were actually using the unnormalized DSM objective-based model [1] to calculate the score norm. Then, yes, the model's output could contain some information regarding the norm of the score. Still, I must point out that nowadays, this particular objective is no longer used, even in their follow-up [2]. This is certainly the case for [3] (which the authors claimed to use in later parts of the response? I feel there is a contradiction here, so I ask the authors to really make it clear how exactly this is done), and [4]. However, it is clear that this is an established empirical observation [5], so I will not fight against its validity. I feel there might be something else at play, as [5] also used a **normalized training objective.** Again, I certainly feel that **this part needs to be written more clearly,** because I feel like even in the author's response there are some contradictions: e.g. is it using [1] or [3]?
> > >
> > > **Reply**: Thanks for your question! We are sorry for the confusions, **we directly use a time-dependent score network $s_\theta(\mathbf{x}_t, t)$ in [3], which refers to the implementations in [6].** It appears that the neural network trainined within DDPM or score SDE are both eligible to estimate the score information [3][6]. In [1], we know the training objective of Denoising Score Matching (DSM) is to minimize:
> > >
> > > $$\frac{1}{2} \mathbb{E} _{q _\sigma(\tilde{\mathbf{x}} | \mathbf{x}) p _{\text{data}}(\mathbf{x})} \left[ \left\| \mathbf{s} _\theta(\tilde{\mathbf{x}}) - \nabla _{\tilde{\mathbf{x}}} \log q _\sigma(\tilde{\mathbf{x}} | \mathbf{x}) \right\| _2^2 \right],$$
> > >
> > > the NCSN is then utilized in [5] only for purification and [5] points out that the score norm computed from NCSN is valuable for adversarial detection. This early work [5] provide valuable insights but the NCSN is improved by later SMLD [7] and now **depreciated** just like you said. In [6], the authors compute the scores of input $\mathbf{x} _t$ at different noise level $t$ by using the score network in [3] and [8] directly, for CIFAR-10 and ImageNet-1K respectively. In [3], the authors train a time-dependent score network $s _\theta(\mathbf{x} _t, t)$ by minimizing:
> > >
> > > $$\mathbb{E} _{t, \mathbf{x} _t, \mathbf{x}} \left[ \lambda(t) \left\| \mathbf{s} _\theta(\mathbf{x} _t, t) - \nabla _{\mathbf{x} _t} \log p _t(\mathbf{x} _t | \mathbf{x}) \right\| _2^2 \right],$$
> > >
> > > which is a continous generalization of objective in [1]. Here the main difference is the $\lambda(t)$ is introduced into the objective, which is a positive weight function for continous-time training. The training objective of Score SDE is to ensure that the model learns the score across the entire time continuum, rather than only at discrete noise levels $\sigma$ in [1]. **The noise perturbations used in SMLD [7] and DDPM can be regarded as discretizations of two different SDEs [3].** Hence, **the EPS computation is eligible for both pretrained DDPM and Score SDE diffusion models**, the experiment conducted in [6] utilized both DDPM and Score SDE for CIFAR-10 and ImageNet-1K, respectively. Table 1 provides straight comparison between two models utilized in ADP [5] and Diffusion-based purification, DiffPure [9].
> > >
> > > Table1: Comparing NCSN with Score SDE in terms of noise levels $t$ and training objectives.
> > > |Comparison|NCSN[1]|Score SDE[3]|
> > > |--|--|--|
> > > |**Noise Scale**|finite discrete noise levels $(\sigma_1,\sigma_2,...)$|cntinous noise levels $t$, $t \in [0,1]$|
> > > |**Training Objective**|Denoising Score Matching|Continous Time-dependent Score Matching|
> > > |**Score Network**|$s_\theta(\mathbf{x}, \sigma)$|$s_\theta(\mathbf{x}_t, t)$|
> > >
> > > [1] Generative Modeling by Estimating Gradients of The Data Distribution, NeurIPS 2019.
> > >
> > > [2] Improved techniques for training score-based generative models, NeurIPS 2020.
> > >
> > > [3] Score-Based Generative Modeling Through Stochastic Differential Equations, ICLR 2021.
> > >
> > > [4] Elucidating the Design Space of Diffusion-Based Generative Models. NeurIPS 2022.
> > >
> > > [5] Adversarial Purification with Score-Based Generative Models, ICML 2021.
> > >
> > > [6] Detecting Adversarial Data by Probing Multiple Perturbations Using Expected Perturbation Score, ICML 2023.
> > >
> > > [7] Improved techniques for training score-based generative models, NeurIPS 2020.
> > >
> > > [8] Denoising Diffusion Probabilistic Models, NeurIPS 2020.
> > >
> > > [9] Diffusion Models for Adversarial Purification, ICML2022.

---

> > > ### Author Response · Authors · 2024-11-27
> > > **Response to Reviewer 2MWZ further concerns - (2/2)**
> > >
> > > >Q2. Do you **add Gaussian noise to the (clean/adversarial) sample before making that calculation**?
> > >
> > > **Reply**: Thanks for your question! **Yes**, as we utilized score network $s_\theta(\mathbf{x}_t, t)$ in [1], we use perturbed data (noisy data) to compute the corresponding score, with noisy input $\mathbf{x}_t$ and $t$. We would like you to notice that this is an **indenpendent** process to the purification process, which the noise injection here is only for fitting to the pretrained score network and **we no longer use any version of the noisy data in the subsequent process.** The principle of EPS is to aggregate the score information of many $\mathbf{x}_t$s, where $t \sim \mathcal{U}(0, T)$. **After we compute the score/EPS score, we use the score norms to reweight adaptive timesteps for each original input (a.k.a. reweighting function).** Then, we use the original (clean) adversarial/natural data for the diffusion purification process.
> > >
> > > [1] Score-Based Generative Modeling Through Stochastic Differential Equations, ICLR 2021.

---

> > > ### Author Response · Authors · 2024-11-27
> > > **Response to Reviewer 2MWZ**
> > >
> > > In the end, we want to thank you again for providing this valuable feedback to us, which is always important to hear opinions from other experts. If you feel there are still unclear points regarding our paper, please discuss with us in the author-reviewer discussion phase. If you feel that your concerns are well-addressed, we do hope for **an updated confidence or score** and new comments (if possible).^^ Thanks again!

---

> > > > ### Author Response · Authors · 2024-12-02
> > > >
> > > > Dear Reviewer 2MWZ,
> > > >
> > > > Many thanks for your valuable comments again!
> > > >
> > > > Since the current recommendation is borderline instead of "a good paper", we are not sure if there is anything we can do to make our paper even better? We can further strengthen our paper based on your new comments until our paper is good to be accepted. ^^
> > > >
> > > > We also want to share with you the current situation regarding the other reviews. Unfortunately, Reviewer mLed gave a rejection recommendation despite acknowledging that we addressed all the concerns.
> > > >
> > > > This has placed our submission in a particularly challenging position (i.e., the averaged rating is around borderline). We would be deeply grateful for your consideration of further raising your confidence or score. Your support would play a crucial role in ensuring our paper can be fairly evaluated in this rebuttal.
> > > >
> > > > Looking forward to hearing back from you.
> > > >
> > > > Best regards,
> > > >
> > > > Authors of Submission 10115

---

> > > ### Author Response · Authors · 2024-12-03
> > > **Official Comment by Authors**
> > >
> > > Dear Reviewer 2MWZ,
> > >
> > > Many thanks for your valuable comments again!
> > >
> > > Since the current recommendation is borderline instead of "a good paper", we are not sure if there is anything we can do to make our paper even better? We can further strengthen our paper based on your new comments until our paper is good to be accepted. ^^
> > >
> > > We also want to share with you the current situation regarding the other reviews. Unfortunately, Reviewer mLed gave a rejection recommendation despite acknowledging that we addressed all the concerns.
> > >
> > > This has placed our submission in a particularly challenging position (i.e., the averaged rating is around borderline). We would be deeply grateful for your consideration of further raising your confidence or score. Your support would play a crucial role in ensuring our paper can be fairly evaluated in this rebuttal.
> > >
> > > Looking forward to hearing back from you.
> > >
> > > Best regards,
> > >
> > > Authors of Submission 10115

---

### Author Response · Authors · 2024-12-04
**Revision Update and Summary of Changes (part 1/2)**

We appreciate the thorough feedback from all reviewers and have addressed each concern through additional experiments and clarifications.

We outline our key responses and improvements to the manuscript in this summary.

**Responses to Reviewer 2MWZ**
1. **Comprehensive background introduction**
- Concerns: The reviewer stated that **"I would like to make a note that my background is more on diffusion rather than adversarial perturbation."**
- Response: We **identified potential misunderstandings** due to insufficient background information on Diffusion-Based Purification (DBP) and adversarial perturbations provided in the **Preliminary (Section 2)**. We **clarify the problem settings** in diffusion-based generative tasks versus adversarial classification and **outline the high-level objective, motivation, research gap and contributions** of this study, ensuring that readers from diverse backgrounds can fully understand the context.

2. **Observation in Figure 4**
- Concerns: What exactly the authors observe here in Figure 4 in Appendix B?
- Responses: We explained that the y-axis represents the EPS score norm and clarified that **"with increased attack strength, the score norms also increase."**. We clarify that our study **leverages the observation** that data farther from the original distribution tends to exhibit greater momentum (score norm) towards high-density areas in the vector field, as supported by [1].

[1] Generative Modeling by Estimating Gradients of The Data Distribution, NeurIPS 2019.

3. **Score Norm Computation**
- Concerns: How is this score norm calculated?
- Responses: We **clarified** that the score is estimated using the pre-trained score network based on **score matching principles**, not directly computed from the diffusion model. The norm is then derived mathematically from these estimated scores.

4. **Noise Injection**
- Concerns: Why is a proportional amount of noise added again to the sample?
- Response: We **clarified** by distinguishing adversarial perturbations from the Gaussian noise injected during diffusion. We emphasized that adding Gaussian noise to adversarial examples is **a well-established defense mechanism** in DBP.

5. **Timestep Setting**
- Concerns: Algorithm 1 essentially forwards the sample to $t+t’$ time, right? Also, now when you run the backward sampling process, do you start at $t+t’$ or $t’$?
- Responses: We clarified the **potential misunderstanding** that no additional $t^{\prime}$ is introduced. Instead, we adapt the optimal timestep $t^*$ based on the sample’s score norm. We have **revised** Algorithm 1’s notation for better readability.

6. **Test Sample**
- Concerns: Are you treating the test (start) sample x as a clean or a noisy one in terms of diffusion?
- Responses: We **clarified** and **explained** that test samples are treated uniformly $(t=0)$ regardless of whether they are adversarial or natural.

7. **Bias Term**
- Concerns: How is the bias term b selected?
- Responses: We justified the reweighting functions between **lines 336–363** of the revision and conducted an ablation study on the bias term $b$. We appended these results to **Appendix J**.

8. **Function Evaluation**
- Concerns: Are you suggesting to evaluate the pre-trained diffusion model on many ts? If so, this requires many network evaluations, right?
- Responses: We have **corrected** the unclear definition and clarified that EPS computation **does not involve repeated evaluations** of the diffusion model.

10. **Other Minor Updates**
- We **simplified** the caption of Figure 1 as suggested.

---

> ### Author Response · Authors · 2024-12-04
> **Revision Update and Summary of Changes (part 2/2)**
>
> **Responses to Reviewer LRdc**
> 1. **Experiment Metric**
> - Concerns: How is the standard accuracy calculated?
> - Responses: We **explained** that standard accuracy refers to performance under natural data evaluation and is critical for understanding the trade-off between clean accuracy and robustness.
>
> 2. **Purification Process**
> - Concerns: Are the clean samples also purified before applying the pre-trained classifier network?
> - Responses: We **clarified** and **explained** that test samples are treated uniformly $(t=0)$ regardless of whether they are adversarial or natural.
>
> 3. **EPS Configuration**
> - Concerns: What is the value of $t$ within $s_\theta$ in eq (5)?
> - Responses: We **detailed** hyperparamter settings for EPS computation in **lines 332-335**.
>
> 4. **Connection between Prop 1 and Motivation**
> - Concerns: How does the theoretical result in sec 3 support the claimed relationship between score norm and the optimal noise level?
> - Responses: We **revise** and **reorganize** Section 3 to clarify the connection between Prop 1, adversarial perturbation budget, and optimal denoise efforts (noise level). Prop 1 establishes a relationship between **different samples' score norms and noise levels** and motivates the adoption of score norms to reweight $t^*$ and align the purification process with the sample-specific perturbation budget, based on each sample's score norm.
> We have also lowered the claim of theoretical contribution of Prop 1 and **highlighted** the contribution as the proposed SSNI framework itself.
>
> **Responses to Reviewer mLed**
> 1. **Ablation Study**
> - Concerns: How to adapt bias in the reweighting function in different datasets and networks?
> - Responses: We **supplemented experiments** to demonstrate the adaptability of the reweighting function across multiple classifiers (WRN-28-10, WRN-70-16 on CIFAR-10, and RN-50 on ImageNet-1K) in **Appendix J**.
>
> **Responses to Reviewer iJtf**
> 1. **Discussion of Relevant Literature**
> - Concerns: How does the study demonstrate novelty compared to the reference?
> - Responses: We **append the discussion** and **highlight the key difference** between our study and the referenced study **in Section 2**.
>
> 2. **Elaboration on Derivations**:
> - Concerns: Elaborate some intermediate steps of Lemma 3 and Proposition 1.
> - Responses: We **revise Appendix A** to fully detail the intermediate steps in the proofs of Lemma 3 and Proposition 1. Specifically:
>     - We update the condition under which Lemma 3 holds;
>     - We clarify the use of key tools, e.g., triangle inequality in conjunction with the Cauchy-Schwarz inequality, and the continuous approximation and first-order Taylor expansion when deriving the derivative of $\bar{\alpha}_t$.
>
>  3. **Attack Types**
>  - Concerns: Why BPDA+EOT is chosen to evaluate the proposed diffusion-based purification method?
>  - Responses: We clarified that our main **Experiment Results** are evaluated under the **Golden-standard** of DBP methods, which is PGD+EOT white-box adaptive attack. Also, we would like to comprehensively evaluate our proposed framework under diverse attack types, where BPDA+EOT is significantly different from PGD+EOT attack in the fundamental principle.
>
>  ---
> We believe these responses have comprehensively addressed **all reviewers' concerns**, including follow-up questions raised by reviewer LRdc.
> The revision incorporates additional experiments, clarifications, and corrected derivations.
> The reviewer (mLed) acknowledged that we addressed all the concerns.
> The reviewers (2MWZ, iJtf) have not raised further issues and have indicated scores leaning toward acceptance.
>
> Once again, we appreciate all reviewers' efforts and time, which have helped us to enrich the quality and clarity of our submission.

---

### Meta-Review · Area_Chair_vBwD · 2024-12-22

**Metareview:**

The paper presents a refinement of the DBP framework by tailoring noise levels based on the samples. The work is well motivated as the original DBP had kept the question of suitable noise level open. The reviewers mostly agree that the work makes a reasonable advance, however there are quite a few concerns regarding the work including details regarding score norm scale, limited clarity in some aspects of the technical exposition, and aspects of the empirical evaluation. There is also literature on attacking DBP, which is not considered/cited in the work and is a major concern, e.g., see King et al., DiffAttack: Evasion Attacks Against Diffusion-Based Adversarial Purification, NeurIPS'23. To their credit, the authors have addressed some of the concerns from the reviewers, e.g., norm scaling, etc., but other concerns persist.

**Additional Comments On Reviewer Discussion:**

The reviewers engaged with the authors during discussion period and the discussions led to improved clarity on the work.

---

### Decision · Program_Chairs · 2025-01-22

Reject